# Model Fusion via Optimal Transport

**Sidak Pal Singh**[*]
ETH Zurich, Switzerland
contact@sidakpal.com

**Martin Jaggi**
EPFL, Switzerland
martin.jaggi@epfl.ch

## Abstract

Combining different models is a widely used paradigm in machine learning applications. While the most common approach is to form an ensemble of models and average their individual predictions, this approach is often rendered infeasible by given resource constraints in terms of memory and computation, which grow linearly with the number of models. We present a layer-wise model fusion algorithm for neural networks that utilizes optimal transport to (soft-) align neurons across the models before averaging their associated parameters.

We show that this can successfully yield "one-shot" knowledge transfer (i.e, without requiring any retraining) between neural networks trained on heterogeneous non-i.i.d. data. In both i.i.d. and non-i.i.d. settings, we illustrate that our approach significantly outperforms vanilla averaging, as well as how it can serve as an efficient replacement for the ensemble with moderate fine-tuning, for standard convolutional networks (like VGG11), residual networks (like RESNET18), and multi-layer perceptrons on CIFAR10, CIFAR100, and MNIST. Finally, our approach also provides a principled way to combine the parameters of neural networks with different widths, and we explore its application for model compression. The code is available at the following link, https://github.com/sidak/otfusion.

## 1 Introduction

If two neural networks had a child, what would be its weights? In this work, we study the fusion of two *parent* neural networks—which were trained differently but have the same number of layers—into a single *child* network. We further focus on performing this operation in a *one-shot manner*, based on the network weights only, so as to minimize the need of any retraining.

This fundamental operation of merging several neural networks into one contrasts other widely used techniques for combining machine learning models:

*Ensemble methods* have a very long history. They combine the outputs of several different models as a way to improve the prediction performance and robustness. However, this requires maintaining the $K$ trained models and running each of them at test time (say, in order to average their outputs). This approach thus quickly becomes infeasible for many applications with limited computational resources, especially in view of the ever-growing size of modern deep learning models.

The simplest way to fuse several parent networks into a single network of the same size is direct *weight averaging*, which we refer to as vanilla averaging; here for simplicity, we assume that all network architectures are identical. Unfortunately, neural networks are typically highly redundant in their parameterizations, so that there is no one-to-one correspondence between the weights of two different neural networks, even if they would describe the same function of the input. In practice, vanilla averaging is known to perform very poorly on trained networks whose weights differ non-trivially.

Finally, a third way to combine two models is *distillation*, where one network is retrained on its training data, while jointly using the output predictions of the other 'teacher' network on those

---

[*]Work done while at EPFL.

samples. Such a scenario is considered infeasible in our setting, as we aim for approaches not requiring the sharing of training data.This requirement is particularly crucial if the training data is to be kept private, like in federated learning applications, or is unavailable due to e.g. legal reasons.

**Contributions.** We propose a novel layer-wise approach of aligning the neurons and weights of several differently trained models, for fusing them into a single model of the same architecture. Our method relies on optimal transport (OT) [1, 2], to minimize the transportation cost of neurons present in the layers of individual models, measured by the similarity of activations or incoming weights. The resulting layer-wise averaging scheme can be interpreted as computing the Wasserstein barycenter [3, 4] of the probability measures defined at the corresponding layers of the parent models.

We empirically demonstrate that our method succeeds in the one-shot merging of networks of different weights, and in all scenarios significantly outperforms vanilla averaging. More surprisingly, we also show that our method succeeds in merging two networks that were trained for slightly different tasks (such as using a different set of labels). The method is able to "inherit" abilities unique to one of the parent networks, while outperforming the same parent network on the task associated with the other network. Further, we illustrate how it can serve as a data-free and algorithm independent post-processing tool for structured pruning. Finally, we show that OT fusion, with mild fine-tuning, can act as efficient proxy for the ensemble, whereas vanilla averaging fails for more than two models.

**Extensions and Applications.** The method serves as a new building block for enabling several use-cases: (1) The adaptation of a global model to personal training data. (2) Fusing the parameters of a bigger model into a smaller sized model and vice versa. (3) Federated or decentralized learning applications, where training data can not be shared due to privacy reasons or simply due to its large size. In general, improved model fusion techniques such as ours have strong potential towards encouraging model exchange as opposed to data exchange, to improve privacy & reduce communication costs.

## 2   Related Work

**Ensembling.** Ensemble methods [5–7] have long been in use in deep learning and machine learning in general. However, given our goal is to obtain a single model, it is assumed infeasible to maintain and run several trained models as needed here.

**Distillation.** Another line of work by Hinton et al. [8], Buciluǎ et al. [9], Schmidhuber [10] proposes distillation techniques. Here the key idea is to employ the knowledge of a pre-trained teacher network (typically larger and expensive to train) and transfer its abilities to a smaller model called the student network. During this transfer process, the goal is to use the relative probabilities of misclassification of the teacher as a more informative training signal.

While distillation also results in a single model, the main drawback is its computational complexity— the distillation process is essentially as expensive as training the student network from scratch, and also involves its own set of hyper-parameter tuning. In addition, distillation still requires sharing the training data with the teacher (as the teacher network can be too large to share), which we avoid here.

In a different line of work, Shen et al. [11] propose an approach where the student network is forced to produce outputs mimicking the teacher networks, by utilizing Generative Adversarial Network [12]. This still does not resolve the problem of high computational costs involved in this kind of knowledge transfer. Further, it does not provide a principled way to aggregate the parameters of different models.

**Relation to other network fusion methods.** Several studies have investigated a method to merge two trained networks into a single network without the need for retraining [13–15]. Leontev et al. [15] propose Elastic Weight Consolidation, which formulates an assignment problem on top of diagonal approximations to the Hessian matrices of each of the two parent neural networks. Their method however only works when the weights of the parent models are already close, i.e. share a significant part of the training history [13, 14], by relying on SGD with periodic averaging, also called local SGD [16]. Nevertheless, their empirical results [15] do not improve over vanilla averaging.

**Alignment-based methods.** Alignment of neurons was considered in Li et al. [17] to probe the representations learned by different networks. Recently, Yurochkin et al. [18] independently proposed a Bayesian non-parametric framework that considers matching the neurons of different MLPs in federated learning. In a concurrent work[2], Wang et al. [19] extend [18] to more realistic networks

including CNNs, also with a specific focus on federated learning. In contrast, we develop our method from the lens of optimal transport (OT), which lends us a simpler approach by utilizing Wasserstein barycenters. The method of aligning neurons employed in both lines of work form instances for the choice of ground metric in OT. Overall, we consider model fusion in general, beyond federated learning. For instance, we show applications of fusing different sized models (e.g., for structured pruning) as well as the compatibility of our method to serve as an initialization for distillation. From a practical side, our approach is # of layer times more efficient and also applies to ResNets.

To conclude, the application of Wasserstein barycenters for averaging the weights of neural networks has—to our knowledge—not been considered in the past.

## 3 Background on Optimal Transport (OT)

We present a short background on OT in the discrete case, and in this process set up the notation for the rest of the paper. OT gives a way to compare two probability distributions defined over a ground space $\mathcal{S}$, provided an underlying distance or more generally the cost of transporting one point to another in the ground space. Next, we describe the linear program (LP) which lies at the heart of OT.

**LP Formulation.** First, let us consider two empirical probability measures $\mu$ and $\nu$ denoted by a weighted sum of Diracs, i.e., $\mu = \sum_{i=1}^{n} \alpha_i\, \delta(\boldsymbol{x}^{(i)})$ and $\nu = \sum_{i=1}^{m} \beta_i\, \delta(\boldsymbol{y}^{(i)})$. Here $\delta(\boldsymbol{x})$ denotes the Dirac (unit mass) distribution at point $\boldsymbol{x} \in \mathcal{S}$ and the set of points $\boldsymbol{X} = (\boldsymbol{x}^{(1)}, \ldots, \boldsymbol{x}^{(n)}) \in \mathcal{S}^n$. The weight $\boldsymbol{\alpha} = (\alpha_1, \ldots, \alpha_n)$ lives in the probability simplex (and similarly $\boldsymbol{\beta}$). Further, let $\boldsymbol{C}_{ij}$ denote the ground cost of moving point $\boldsymbol{x}^{(i)}$ to $\boldsymbol{y}^{(j)}$. Then the optimal transport between $\mu$ and $\nu$ can be formulated as solving the following linear program. $\mathrm{OT}(\mu, \nu; \boldsymbol{C}) := \min \langle \boldsymbol{T}, \boldsymbol{C} \rangle$, with $\boldsymbol{T} \in \mathbb{R}_{+}^{(n \times m)}$ such that $\boldsymbol{T}\boldsymbol{1}_m = \boldsymbol{\alpha}$, $\boldsymbol{T}^{\top}\boldsymbol{1}_n = \boldsymbol{\beta}$. Here, $\langle \boldsymbol{T}, \boldsymbol{C} \rangle := \mathrm{tr}\left(\boldsymbol{T}^{\top}\boldsymbol{C}\right) = \sum_{ij} T_{ij} C_{ij}$ is the Frobenius inner product of matrices. The optimal $\boldsymbol{T} \in \mathbb{R}_{+}^{(n \times m)}$ is called as the *transportation matrix* or *transport map*, and $T_{ij}$ represents the optimal amount of mass to be moved from point $\boldsymbol{x}^{(i)}$ to $\boldsymbol{y}^{(j)}$.

**Wasserstein Distance.** When $\mathcal{S} = \mathbb{R}^d$ and the cost is defined with respect to a metric $D_{\mathcal{S}}$ over $\mathcal{S}$ $\left(\text{i.e., } C_{ij} = D_{\mathcal{S}}(\boldsymbol{x}^{(i)}, \boldsymbol{y}^{(j)})^p \text{ for any } i, j\right)$, OT establishes a distance between probability distributions. This is called the $p$-Wasserstein distance and is defined as $\mathcal{W}_p(\mu, \nu) := \mathrm{OT}(\mu, \nu; D_{\mathcal{S}}^p)^{1/p}$.

**Wasserstein Barycenters.** This represents the notion of averaging in the Wasserstein space. To be precise, the Wasserstein barycenter [3] is a probability measure that minimizes the weighted sum of ($p$-th power) Wasserstein distances to the given $K$ measures $\{\mu_1, \ldots, \mu_K\}$, with corresponding weights $\boldsymbol{\eta} = \{\eta_1, \ldots, \eta_K\} \in \Sigma_K$. Hence, it can be written as $\mathcal{B}_p(\mu_1, \ldots, \mu_K) = \arg\min_{\mu} \sum_{k=1}^{K} \eta_k\, \mathcal{W}_p(\mu_k, \nu)^p$.

## 4 Proposed Algorithm

In this section, we discuss our proposed algorithm for model aggregation. First, we consider that we are averaging the parameters of only two neural networks, but later present the extension to the multiple model case. For now, we ignore the bias parameters and we only focus on the weights. This is to make the presentation succinct, and it can be easily extended to take care of these aspects.

**Motivation.** As alluded to earlier in the introduction, the problem with vanilla averaging of parameters is the lack of one-to-one correspondence between the model parameters. In particular, for a given layer, there is no direct matching between the neurons of the two models. For e.g., this means that the $p^{\text{th}}$ neuron of model A might behave very differently (in terms of the feature it detects) from the $p^{\text{th}}$ neuron of the other model B, and instead might be quite similar in functionality to the $p + 1^{\text{th}}$ neuron. Imagine, if we knew a perfect matching between the neurons, then we could simply align the neurons of model A with respect to B. Having done this, it would then make more sense to perform vanilla averaging of the neuron parameters. The matching or assignment could be formulated as a permutation matrix, and just multiplying the parameters by this matrix would align the parameters.

But in practice, it is more likely to have soft correspondences between the neurons of the two models for a given layer, especially if their number is not the same across the two models. This is where optimal transport comes in and provides us a soft-alignment matrix in the form of the transport map $\boldsymbol{T}$. In other words, the alignment problem can be rephrased as optimally transporting the neurons in a given layer of model A to the neurons in the same layer of model B.

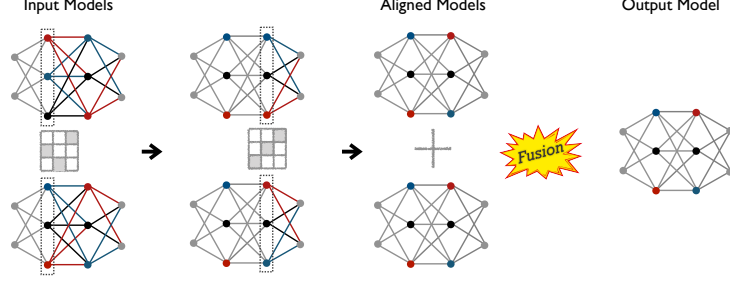

Figure 1: **Model Fusion procedure**: The first two steps illustrate how the model A (top) gets aligned with respect to model B (bottom). The alignment here is reflected by the ordering of the node colors in a layer. Once each layer has been aligned, the model parameters get averaged (shown by the +).

**General procedure.** Let us assume we are at some layer $\ell$ and that neurons in the previous layers have already been aligned. Then, we define probability measures over neurons in this layer for the two models as, $\mu^{(\ell)} = \big(\boldsymbol{\alpha}^{(\ell)}, \boldsymbol{X}[\ell]\big)$ and $\nu^{(\ell)} = \big(\boldsymbol{\beta}^{(\ell)}, \boldsymbol{Y}[\ell]\big)$, where $\boldsymbol{X}, \boldsymbol{Y}$ are the measure supports.

Next, we use uniform distributions to initialize the histogram (or probability mass values) for each layer. Although we note that it is possible to additionally use other measures of neuron importance [20, 21], but we leave it for a future work. In particular, if the size of layer $\ell$ of models A and B is denoted by $n^{(\ell)}, m^{(\ell)}$ respectively, we get $\boldsymbol{\alpha}^{(\ell)} \leftarrow \mathbf{1}_{n^{(\ell)}}/n^{(\ell)}$, $\boldsymbol{\beta}^{(\ell)} \leftarrow \mathbf{1}_{m^{(\ell)}}/m^{(\ell)}$. Now, in terms of the alignment procedure, we first align the incoming edge weights for the current layer $\ell$. This can be done by post-multiplying with the previous layer transport matrix $\boldsymbol{T}^{(\ell-1)}$, normalized appropriately via the inverse of the corresponding column marginals $\boldsymbol{\beta}^{(\ell-1)}$:

$$\widehat{\boldsymbol{W}}_A^{(\ell,\,\ell-1)} \leftarrow \boldsymbol{W}_A^{(\ell,\,\ell-1)} \boldsymbol{T}^{(\ell-1)} \text{diag}\big(1/\boldsymbol{\beta}^{(\ell-1)}\big). \tag{1}$$

This update can be interpreted as follows: the matrix $\boldsymbol{T}^{(\ell-1)} \text{diag}\big(\boldsymbol{\beta}^{-(\ell-1)}\big)$ has $m^{(\ell-1)}$ columns in the simplex $\Sigma_{n^{(\ell-1)}}$, thus post-multiplying $\boldsymbol{W}_A^{(\ell,\,\ell-1)}$ with it will produce a convex combination of the points in $\boldsymbol{W}_A^{(\ell,\,\ell-1)}$ with weights defined by the optimal transport map $\boldsymbol{T}^{(\ell-1)}$.

Once this has been done, we focus on aligning the neurons in this layer $\ell$ of the two models. Let us assume, we have a suitable ground metric $D_{\mathcal{S}}$ (which we discuss in the sections ahead). Then we compute the optimal transport map $\boldsymbol{T}^{(\ell)}$ between the measures $\mu^{(\ell)}, \nu^{(\ell)}$ for layer $\ell$, i.e., $\boldsymbol{T}^{(\ell)}, \mathcal{W}_2 \leftarrow \text{OT}(\mu^{(\ell)}, \nu^{(\ell)}, D_{\mathcal{S}})$, where $\mathcal{W}_2$ denotes the obtained Wasserstein-distance. Now, we use this transport map $\boldsymbol{T}^{(\ell)}$ to align the neurons (more precisely the weights) of the first model (A) with respect to the second (B),

$$\widetilde{\boldsymbol{W}}_A^{(\ell,\,\ell-1)} \leftarrow \text{diag}\big(1/\boldsymbol{\beta}^{(\ell)}\big) {\boldsymbol{T}^{(\ell)}}^{\top} \widehat{\boldsymbol{W}}_A^{(\ell,\,\ell-1)}. \tag{2}$$

We will refer to model A's weights, $\widetilde{\boldsymbol{W}}_A^{(\ell,\,\ell-1)}$, as those aligned with respect to model B. Hence, with this alignment in place, we can average the weights of two layers to obtain the fused weight matrix $\boldsymbol{W}_{\mathcal{F}}^{(\ell,\,\ell-1)}$, as in Eq. (3). We carry out this procedure over all the layers sequentially.

$$\boldsymbol{W}_{\mathcal{F}}^{(\ell,\,\ell-1)} \leftarrow \tfrac{1}{2}\big(\widetilde{\boldsymbol{W}}_A^{(\ell,\,\ell-1)} + \boldsymbol{W}_B^{(\ell,\,\ell-1)}\big). \tag{3}$$

Note that, since the input layer is ordered identically for both models, we start the alignment from second layer onwards. Additionally, the order of neurons for the very last layer, i.e., in the output layer, again is identical. Thus, the (scaled) transport map at the last layer will be equal to the identity.

**Extension to multiple models.** The key idea is to begin with an estimate $\widehat{M}_{\mathcal{F}}$ of the fused model, then align all the given models with respect to it, and finally return the average of these aligned weights as the final weights for the fused model. For the two model case, this is equivalent to the procedure we discussed above when the fused model is initialized to model B, i.e., $\widehat{M}_{\mathcal{F}} \leftarrow M_B$. Because, aligning model B with this estimate of the fused model will yield a (scaled) transport map equal to the identity. And then, Eq. (3) will amount to returning the average of the aligned weights.

**Alignment strategies.** The above discussion implies that we need to design a ground metric $D_{\mathcal{S}}$ between the inter-model neurons. So, we branch out into the following two strategies:

*(a) Activation-based alignment* ($\psi = $ 'acts'): In this variant, we run inference over a set of $m$ samples, $S = \{\mathbf{x}\}_{i=1}^{m}$ and store the activations for all neurons in the model. Thus, we consider the neuron activations, concatenated over the samples into a vector, as the support of the measures, and we denote it as $\boldsymbol{X}_k \leftarrow \text{ACTS}\big(M_k(S)\big)$, $\boldsymbol{Y} \leftarrow \text{ACTS}\big(M_{\mathcal{F}}(S)\big)$. Then the neurons across the two models are considered to be similar if they produce similar activation outputs for the given set of samples. We measure this by computing the Euclidean distance between the resulting vector of activations. This serves as the ground metric for OT computations. In practice, we use the pre-activations.

*(b) Weight-based alignment* ($\psi = $ 'wts'): Here, we consider that the support of each neuron is given by the weights of the incoming edges (stacked in a vector). Thus, a neuron can be thought as being represented by the row corresponding to it in the weight matrix. So, the support of the measures in such an alignment type is given by, $\boldsymbol{X}_k[\ell] \leftarrow \widehat{\boldsymbol{W}}_k^{(\ell, \ell-1)}$, $\boldsymbol{Y}[\ell] \leftarrow \widehat{\boldsymbol{W}}_{\mathcal{F}}^{(\ell, \ell-1)}$. The reasoning for such a choice for the support stems from the neuron activation at a particular layer being calculated as the inner product between this weight vector and the previous layer output. The ground metric used for OT is the Euclidean distance, like in the previous alignment strategy. Besides this difference of employing the actual weights in the ground metric (LINE 6, 10), rest of the procedure is identical.

Lastly, the overall procedure is summarized in Algorithm 1 below, where the GETSUPPORT selects between the above strategies based on the value of $\psi$.

---

**Algorithm 1: Model Fusion (with $\psi = \{$'acts', 'wts'$\}-$alignment)**

---

1: **input:**     Trained models $\{M_k\}_{k=1}^{K}$ and initial estimate of the fused model $\widehat{M}_{\mathcal{F}}$

2: **output:**     Fused model $M_{\mathcal{F}}$ with weights $\boldsymbol{W}_{\mathcal{F}}$

3: **notation:**     For model $M_k$, size of the layer $\ell$ is written as $n_k^{(\ell)}$, and the weight matrix between the layer $\ell$ and $\ell - 1$ is denoted as $\boldsymbol{W}_k^{(\ell, \ell-1)}$. Neuron support tensors are given by $\boldsymbol{X}_k, \boldsymbol{Y}$.

4: **initialize:**     The size of input layer $n_k^{(1)} \leftarrow m^{(1)}$ for all $k \in [K]$; so $\boldsymbol{\alpha}_k^{(1)} = \boldsymbol{\beta}^{(1)} \leftarrow \mathbf{1}_{m^{(1)}}/m^{(1)}$ and the transport map is defined as $\boldsymbol{T}_k^{(1)} \leftarrow \text{diag}(\boldsymbol{\beta}^{(1)})\ \mathcal{I}_{m^{(1)} \times m^{(1)}}$.

5: **for** each layer $\ell = 2, \ldots, L$ **do**

6:     $\boldsymbol{\beta}^{(\ell)},\ \boldsymbol{Y}[\ell] \qquad \leftarrow \mathbf{1}_{m^{(\ell)}}/m^{(\ell)},\ \text{GETSUPPORT}(\widehat{M}_{\mathcal{F}}, \psi, \ell)$

7:     $\nu^{(\ell)} \qquad\qquad \leftarrow \big(\boldsymbol{\beta}^{(\ell)},\ \boldsymbol{Y}[\ell]\big)$     ▷ Define probability measure for initial fused model $\widehat{M}_{\mathcal{F}}$

8:     **for** each model $k = 1, \ldots, K$ **do**

9:       $\widehat{\boldsymbol{W}}_k^{(\ell, \ell-1)} \qquad \leftarrow \boldsymbol{W}_k^{(\ell, \ell-1)}\boldsymbol{T}_k^{(\ell-1)}\text{diag}\big(\frac{1}{\boldsymbol{\beta}^{(\ell-1)}}\big)$     ▷ Align incoming edges for $M_k$

10:       $\boldsymbol{\alpha}_k^{(\ell)},\ \boldsymbol{X}_k[\ell] \quad \leftarrow \mathbf{1}_{n_k^{(\ell)}}/n_k^{(\ell)},\ \text{GETSUPPORT}(M_k, \psi, \ell)$

11:       $\mu_k^{(\ell)} \qquad\qquad \leftarrow \big(\boldsymbol{\alpha}_k^{(\ell)},\ \boldsymbol{X}_k[\ell]\big)$     ▷ Define probability measure for model $M_k$

12:       $D_{\mathcal{S}}^{(\ell)}[p, q] \qquad \leftarrow \|\boldsymbol{X}_k[\ell][p] - \boldsymbol{Y}[\ell][q]\|_2,\ \forall\, p \in [n_k^{(\ell)}], q \in [m^{(\ell)}]$     ▷ Form ground metric

13:       $\boldsymbol{T}_k^{(\ell)},\ \mathcal{W}_2^{(\ell)} \qquad \leftarrow \text{OT}\big(\mu_k^{(\ell)}, \nu^{(\ell)}, D_{\mathcal{S}}^{(\ell)}\big)$     ▷ Compute OT map and distance

14:       $\widetilde{\boldsymbol{W}}_k^{(\ell, \ell-1)} \qquad \leftarrow \text{diag}\big(\frac{1}{\boldsymbol{\beta}^{(\ell)}}\big)\boldsymbol{T}^{(\ell)\top}\widehat{\boldsymbol{W}}_k^{(\ell, \ell-1)}$     ▷ Align model $M_k$ neurons

15:     **end for**

16:     $\boldsymbol{W}_{\mathcal{F}}^{(\ell, \ell-1)} \qquad\quad \leftarrow \frac{1}{K}\sum_{k=1}^{K}\widetilde{\boldsymbol{W}}_k^{(\ell, \ell-1)}$     ▷ Average model weights

17: **end for**

---

## 4.1 Discussion

**Pros and cons of alignment type.**     An advantage of the weight-based alignment is that it is independent of the dataset samples, making it useful in privacy-constrained scenarios. On the flip side, the activation-based alignment only needs unlabeled data, and an interesting prospect for a future study would be to utilize synthetic data. But, activation-based alignment may help tailor the fusion to certain desired kinds of classes or domains. Fusion results for both are nevertheless similar.

**Combinatorial hardness of the ideal procedure.**     In principle, we should actually search over the space of permutation matrices, jointly across all the layers. But this would be computationally

intractable for models such as deep neural networks, and thus we fuse in a layer-wise manner and in a way have a greedy procedure.

**# of samples used for activation-based alignment.**     We typically consider a mini-batch of $\sim 100$ to $400$ samples for these experiments. Table S2 in the Appendix, shows that effect of increasing this mini-batch size on the fusion performance and we find that even as few as $25$ samples are enough to outperform vanilla averaging.

**Exact OT and runtime efficiency.**     Our fusion procedure is efficient enough for the deep neural networks considered here (VGG11, RESNET18), so we primarily utilize exact OT solvers. While the runtime of exact OT is roughly cubic in the cardinality of the measure supports, it is not an issue for us as this cardinality (which amounts to the network width) is $\leq 600$ for these networks. In general, modern-day neural networks are typically deeper than wide. To give a concrete estimate, the *time taken to fuse six* VGG11 *models is* $\approx 15$ *seconds* on 1 Nvidia V100 GPU (c.f. Section S1.4 for more details). It is possible to further improve the runtime by adopting the entropy-regularized OT [22], but this looses slightly in terms of test accuracy compared to exact OT (c.f. Table S4).

# 5    Experiments

**Outline.**     We first present our results for one-shot fusion when the models are trained on *different data distributions*. Next, in Section 5.2, we consider (one-shot) fusion in the case when model sizes are different (i.e., unequal layer widths to be precise). In fact, this aspect *facilitates a new tool* that can be applied in ways not possible with vanilla averaging. Further on, we focus on the use-case of obtaining an *efficient* replacement for ensembling models in Section 5.3.

**Empirical Details.**     We test our model fusion approach on standard image classification datasets, like CIFAR10 with commonly used convolutional neural networks (CNNs) such as VGG11 [23] and residual networks like ResNet18 [24]; and on MNIST, we use a fully connected network with 3 hidden layers of size $400, 200, 100$, which we refer to as MLPNET. As baselines, we mention the performance of 'prediction' ensembling and 'vanilla' averaging, besides that of individual models. Prediction ensembling refers to keeping all the models and averaging their predictions (output layer scores), and thus reflects in a way the ideal (but unrealistic) performance that we can hope to achieve when fusing into a single model. Vanilla averaging denotes the direct averaging of parameters. All the performance scores are test accuracies. Full experimental details are provided in Appendix S1.1.

## 5.1    Fusion in the setting of heterogeneous data and tasks

We first consider the setting of merging two models A and B, but assume that model A has some special skill or knowledge (say, recognizing an object) which B does not possess. However, B is overall more powerful across the remaining set of skills in comparison to A. The goal of fusion now is to obtain a single model that can gain from the strength of B on overall skills and also acquire the specialized skill possessed by A. Such a scenario can arise e.g. in reinforcement learning where these models are agents that have had different training episodes so far. Another possible use case lies in federated learning [25], where model A is a client application that has been trained to perform well on certain tasks (like personalized keyword prediction) and model B is the server that typically has a strong skill set for a range of tasks (general language model).

The natural constraints in such scenarios are (a) ensuring privacy and (b) minimization communication frequency. This implies that the training examples can not be shared between A and B to respect privacy and a one-shot knowledge transfer is ideally desired, which eliminates e.g., joint training.

At a very abstract level, these scenarios are representative of aggregating models that have been trained on non-i.i.d data distributions. To simulate a heterogeneous data-split, we consider the MNIST digit classification task with MLPNET models, where the unique skill possessed by model A corresponds to recognizing one particular 'personalized' label (say 4), which is unknown to B. Model B contains $90\%$ of the remaining training set (i.e., excluding the label 4), while A has the other $10\%$. Both are trained on their portions of the data for 10 epochs , and other training settings are identical.

Figure 2 illustrates the results for fusing models A and B (in different proportions), both when they have different parameter initializations or when they share the same initialization. OT fusion [3] significantly outperforms the vanilla averaging of their parameters in terms of the overall test accuracy

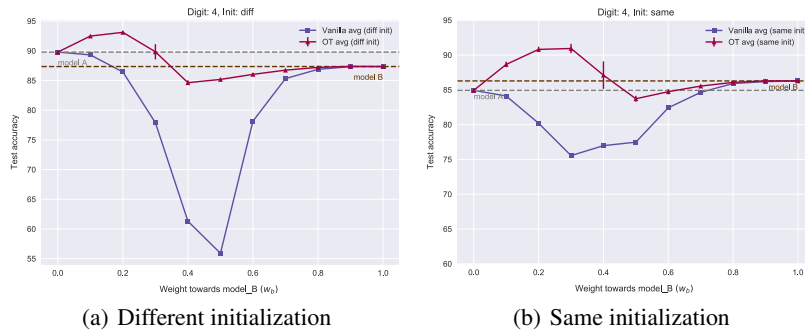

(a) Different initialization      (b) Same initialization

Figure 2: **One-shot skill transfer performance** when the specialist model A and the generalist model B are fused in varying proportions ($w_B$), for different and same initializations. The OT avg. (fusion) curve (in magenta) is obtained by activation-based alignment and we plot the mean performance over 5 seeds along with the error bars for standard deviation. *No retraining is done here.*

in both the cases, and also improves over the individual models. E.g., in Figure 2(a), where the individual models obtain $89.78\%$ and $87.35\%$ accuracy respectively on the overall (global) test set, OT avg. achieves the best overall test set accuracy of $93.11\%$. Thus, confirming the successful skill transfer from both parent models, without the need for any retraining.

Our obtained results are robust to other scenarios when (i) some other label (say 6) serves as the special skill and (ii) the $\%$ of remaining data split is different. These results are collected in the Appendix S5, where in addition we also present results without the special label as well.

**The case of multiple models.** In the above example of two models, one might also consider maintaining an ensemble, however the associated costs for ensembling become prohibitive as soon as the numbers of models increases. Take for instance, four models: A, B, C and D, with the same initialization and assume that A again possessing the knowledge of a special digit (say, 4). Consider that the rest of the data is divided as $10\%, 30\%, 50\%, 10\%$. Now training in the similar setting as before, these models end up getting (global) test accuracies of $87.7\%, 86.5\%, 87.0\%, 83.5\%$ respectively. Ensembling the predictions yields $95.0\%$ while vanilla averaging obtains $80.6\%$. In contrast, OT averaging results in **93.6%** test accuracy ($\approx 6\%$ gain over the best individual model), while being $4\times$ more efficient than ensembling. Further details can be found in the Appendix S7.

## 5.2 Fusing different sized models

An advantage of our OT-based fusion is that it allows the layer widths to be different for each input model. Here, our procedure first identifies which weights of the bigger model should be mapped to the smaller model (via the transport map), and then averages the aligned models (now both of the size of the smaller one). We can thus combine the parameters of a bigger network into a smaller one, and vice versa, allowing new use-cases in (a) model compression and (b) federated learning.

**(a) Post-processing tool for structured pruning.** Structured pruning [26–28] is an approach to model compression that aims to remove entire neurons or channels, resulting in an out-of-the-box reduction in inference costs, while affecting the performance minimally. A widely effective method for CNNs is to remove the filters with smallest $\ell_1$ norm [26]. *Our key idea here is to fuse the original dense network into the pruned network, instead of just throwing it away.*

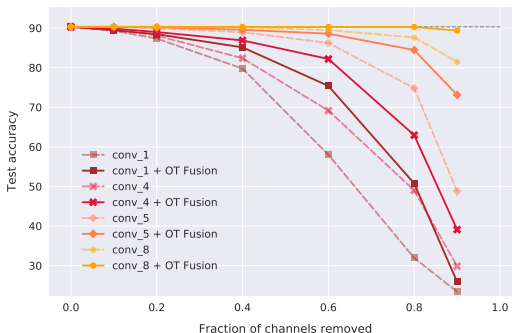

Figure 3: **Post-processing for structured pruning**: Fusing the initial dense VGG11 model into the pruned model helps test accuracy of the pruned model on CIFAR10.

Figure 3 shows the gain in test accuracy on CIFAR10 by carrying out OT fusion procedure (with weight-based alignment) when different convolutional layers of VGG11 are pruned to increasing amounts. For all the layers, we consistently obtain a significant improvement in performance, and $\approx 10\%$ or more gain in the high

sparsity regime. We also observe similar improvements other layers as well as when multiple (or all) layers are pruned simultaneously (c.f. Appendix S8).

Further, these gains are also significant when measured with respect to the overall sparsity obtained in the model. E.g., structured pruning the CONV_8 to $90\%$ results in a net sparsity of $23\%$ in the model. Here after pruning, the accuracy of the model drops from $90.3\%$ to $81.5\%$, and on applying OT fusion, the performances recovers to $89.4\%$. As an another example take CONV_7, where after structured pruning to $80\%$, OT fusion improves the performance of the pruned model from $87.6\%$ to $90.1\%$ while achieving an overall sparsity of $41\%$ in the network (see S8).

Our goal here is not to propose a method for structured pruning, but rather a post-processing tool that can help regain the drop in performance due to pruning. These results are thus independent of the pruning algorithm used, and e.g., Appendix S8 shows similar gains when the filters are pruned based on $\ell_2$ norm (Figure S10) or even randomly (Figure S11). Further, Figure S12 in the appendix also shows the results when applied to VGG11 trained on CIFAR100 (instead of CIFAR10). Overall, OT fusion offers a *completely data-free approach* to improving the performance of the pruned model, which can be handy in the limited data regime or when retraining is prohibitive.

**(b) Adapting the size of client and server-side models in federated learning.** Given the huge sizes of contemporary neural networks, it is evident that we will not able to fit the same sized model on a client device as would be possible on the server. However, this might come at the cost of reduced performance. Further, the resource constraints might be fairly varied even amongst the clients devices, thus necessitating the flexibility to adapt the model sizes.

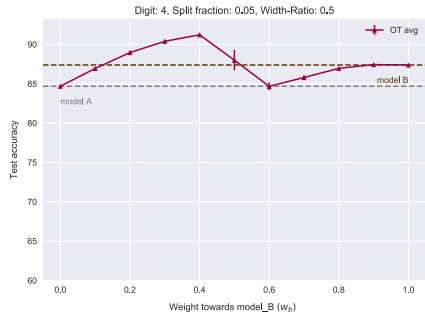

We consider a similar formulation, as in the one-shot knowledge transfer setting from Section 5.1, except that now the model B has twice the layer widths as compared to the corresponding layers of model A. Vanilla averaging of parameters, a core component of the widely prevalent FedAvg algorithm [25], gets ruled out in such a setting. Figure 4 shows how OT fusion/average can still lead to a successful knowledge transfer between the given models.

Figure 4: **One-shot skill transfer for different sized models**: Results of fusing the small client model A into the larger server model B, for varying proportions $w_B$ in which they are fused. See Appendix S6 for more details.

## 5.3 Fusion for efficient ensembling

In this section, our goal is to obtain a single model which can serve as a proxy for an ensemble of models, even if it comes at a slight decrease in performance relative to the ensemble, *for future efficiency*. Specifically, here we investigate how much can be gained by fusing multiple models that differ only in their parameter initializations (i.e., seeds). This means that models are trained on the same data, so unlike in Section 5.1 with a heterogeneous data-split, the gain here might be limited.

We study this in context of deep networks such as VGG11 and RESNET18 which have been trained to convergence on CIFAR10. As a first step, we consider the setting when we are given just two models, the results for which are present in Table 1. We observe that vanilla averaging absolutely fails in this case, and is 3-5× worse than OT averaging, in case of RESNET18 and VGG11 respectively. OT average, however, does not yet improve over the individual models. This can be attributed to the combinatorial hardness of

| DATASET + MODEL | $M_A$ | $M_B$ | PREDICTION AVG. | VANILLA AVG. | OT AVG. | FINETUNING VANILLA | OT |
|---|---|---|---|---|---|---|---|
| CIFAR10 + VGG11 | 90.31 | 90.50 | 91.34 | 17.02 | 85.98 | 90.39 | **90.73** |
| | 1 × | | 1 × | 2 × | 2 × | 2 × | **2 ×** |
| CIFAR10 + RESNET18 | 93.11 | 93.20 | 93.89 | 18.49 | 77.00 | 93.49 | **93.78** |
| | 1 × | | 1 × | 2 × | 2 × | 2 × | **2 ×** |

Table 1: Results for fusing convolutional & residual networks, along with the effect of finetuning the fused models, on CIFAR10. The number below the test accuracies indicate the factor by which a fusion technique is efficient over maintaining all the given models.

the underlying alignment problem, and the greedy nature of our algorithm as mentioned before. As a simple but effective remedy, we consider finetuning (i.e., retraining) from the fused or averaged models. Retraining helps for both vanilla and OT averaging, but in comparison, the OT averaging

| CIFAR100 + VGG11 | INDIVIDUAL MODELS | PREDICTION AVG. | FINETUNING | |
|---|---|---|---|---|
| | | | VANILLA | OT |
| Accuracy | [62.70, 62.57, 62.50, 62.92] | 66.32 | 4.02 | **64.29± 0.26** |
| Efficiency | 1 × | 1 × | 4 × | **4 ×** |
| Accuracy | [62.70, 62.57, 62.50, 62.92, 62.53, 62.70] | 66.99 | 0.85 | **64.55 ± 0.30** |
| Efficiency | 1 × | 1 × | 6 × | **6 ×** |
| Accuracy | [62.70, 62.57, 62.50, 62.92, 62.53, 62.70, 61.60, 63.20] | 67.28 | 1.00 | **65.05± 0.53** |
| Efficiency | 1 × | 1 × | 8 × | **8 ×** |

Table 2: Efficient alternative to ensembling via OT fusion on **CIFAR100** for VGG11. Vanilla average fails to retrain. Results shown are mean ± std. deviation over **5 seeds**.

results in a better score for both the cases as shown in Table 1. E.g., for RESNET18, OT avg. + finetuning gets almost as good as prediction ensembling on test accuracy.

The finetuning scores for vanilla and OT averaging correspond to their best obtained results, when retrained with several finetuning learning rate schedules for a total of 100 and 120 epochs in case of VGG11and RESNET18 respectively. We also considered finetuning the individual models across these various hyperparameter settings (which of course will be infeasible in practice), but the best accuracy mustered via this attempt for RESNET18 was 93.51, in comparison to 93.78 for OT avg. + finetuning. See Appendix S3 and S4 for detailed results and typical retraining curves.

**More than 2 models.** Now, we discuss the case of more than two models, where the savings in efficiency relative to the ensemble are even higher. As before, we take the case of VGG11 on CIFAR10 and additionally CIFAR100 [4], but now consider $\{4, 6, 8\}-$ such models that have been trained to convergence, each from a different parameter initialization. Table 2 shows the results for this in case of CIFAR100 (results for CIFAR10 are similar and can be found in Table S9).

We find that the performance of vanilla averaging degrades to close-to-random performance, and interestingly even fails to retrain, despite trying numerous settings of optimization hyperparameters (like learning rate and schedules, c.f. Section S3.2). In contrast, OT average performs significantly better even without fine-tuning, and results in a mean test accuracy gain $\sim \{1.4\%, 1.7\%, 2\%\}$ over the best individual models after fine-tuning, in the case of $\{4, 6, 8\}-$ base models respectively. Overall, Tables 1, 2 (also S9) show the importance of aligning the networks via OT before averaging. Further finetuning of the OT fused model, always results in an improvement over the individual models, while being # *models times* more efficient than the ensemble.

**Fusion and Distillation.** For the sake of completeness, we also compare OT fusion, distillation, and their combination, in context of transferring the knowledge of a large pre-trained teacher network into a smaller pre-trained student network. We find that starting the distillation from the OT fused model yields better performance than initializing randomly or with the student model. Further, when averaged across the considered temperature values $= \{20, 10, 8, 4, 1\}$, we observe that distillation of the teacher into random or student network based initialization performs worse than simple OT avg. + finetuning (which also doesn't require doing such a sweep that would be prohibitive for larger models/datasets). These experiments are discussed in detail in Appendix S12. An interesting direction for future work would be to use intermediate OT distances computed during fusion as a means for regularizing or distilling with hidden layers.

## 6  Conclusion

We show that averaging the weights of models, by first doing a layer-wise (soft) alignment of the neurons via optimal transport, can serve as a versatile tool for fusing models in various settings. This results in (a) successful one-shot transfer of knowledge between models without sharing training data, (b) data free and algorithm independent post-processing tool for structured pruning, (c) and more generally, combining parameters of different sized models. Lastly, the OT average when further finetuned, allows for just keeping one model rather than a complete ensemble of models at inference. Future avenues include application in distributed optimization and continual learning, besides extending our current toolkit to fuse models with different number of layers, as well as, fusing generative models like GANs [12] (where ensembling does not make as much sense). The promising empirical results of the presented algorithm, thus warrant attention for further use-cases.

## Broader Impact

Model fusion is a fundamental building block in machine learning, as a way of direct knowledge transfer between trained neural networks. Beyond theoretical interest it can serve a wide range of concrete applications. For instance, collaborative learning schemes such as federated learning are of increasing importance for enabling privacy-preserving training of ML models, as well as a better alignment of each individual's data ownership with the resulting utility from jointly trained machine learning models, especially in applications where data is user-provided and privacy sensitive [29]. Here fusion of several models is a key building block to allow several agents to participate in joint training and knowledge exchange. We propose that a reliable fusion technique can serve as a step towards more broadly enabling privacy-preserving and efficient collaborative learning.

### Acknowledgments

We would like to thank Rémi Flamary, Boris Muzellec, Sebastian Stich and other members of MLO, as well as the anonymous reviewers for their comments and feedback.

## Footnotes

[2]An early version of our paper also appeared at NeurIPS 2019 workshop on OT, arxiv:1910.05653.

[3]Only the receiver A's own examples are used for computing the activations, avoiding the sharing of data.

[4]We simply adapt the VGG11 architecture used for CIFAR10 and train it on CIFAR100 for 300 epochs. Since our focus here was not to obtain best individual models, but rather to investigate the efficacy of fusion.

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
