[Supplementary Material]

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

# Appendix

## Contents

## S1 Technical specifications

### S1.1 Experimental Details

**VGG11 training details.** It is trained by SGD for 300 epochs with an initial learning rate of 0.05, which gets decayed by a factor of 2 after every 30 epochs. Momentum $= 0.9$ and weight decay $= 0.0005$. The batch size used is 128. Checkpointing is done after every epoch and the best performing checkpoint in terms of test accuracy is used as the individual model. The block diagram of VGG11 architecture is shown below for reference.

Figure S1: Block diagram of the VGG11 architecture. Adapted from https://bit.ly/2ksX5Eq.

**MLPNET training details.** This is also trained by SGD at a constant learning rate of 0.01 and momentum $= 0.5$. The batch size used is 64.

**RESNET18 training details.** Again, we use SGD as the optimizer, with an initial learning rate of 0.1, which gets decayed by a factor of 10 at epochs $\{150, 250\}$. In total, we train for 300 epochs and similar to the VGG11 setting we use the best performing checkpoint as the individual model. Other than that, momentum $= 0.9$, weight decay $= 0.0001$, and batch size $= 256$. We skip the batch normalization for the current experiments, however, it can possibly be handled by simply multiplying the batch normalization parameters in a layer by the obtained transport map while aligning the neurons.

**Other details.** *Pre-activations.* The results for the activation-based alignment experiments are based on pre-activation values, which were generally found to perform slightly better than post-activation values.

*Regularization.* The regularization constant used for the activation-based alignment results in Table S2 is 0.05.

*Common details.* The bias of a neuron is set to zero in all of the experiments. It is possible to handle it as a regular weight by keeping the corresponding input as 1, but we leave that for future work.

### S1.2 Combining weights and activations for alignment

The output activation of a neuron over input examples gives a good signal about the presence of features in which the neuron gets activated. Hence, one way to combine this information in the above variant with weight-based alignment is to use them in the probability mass values.

In particular, we can take a mini-batch of samples and store the activations of all the neurons. Then we can use the mean activation as a measure of a neuron's significance. But it might be that some neurons produce very high activations (in absolute terms) irrespective of the kind of input examples. Hence, it might make sense to also look at the standard deviation of activations. Thus, one can combine both these factors into an importance weight for the neuron as follows:

$$\text{importance}_k[2, \cdots, L] = \overline{M_k}([x_1, \cdots, x_d]) \odot \sigma(M_k([x_1, \cdots, x_d])) \tag{4}$$

Here, $M_k$ denotes the $k^{\text{th}}$ model into which we pass the inputs $[x_1, \cdots, x_d]$, $\overline{M}$ denotes the mean, $\sigma(.)$ denotes the standard deviation and $\odot$ denotes the elementwise product. Thus, we can now set the probability mass values $b_k^{(l)} \propto \text{importance}_k[l]$, and the rest of the algorithm remains the same.

### S1.3 Optimal Transport

We make use of the Python Optimal Transport (POT)[S1] for performing the computation of Wasserstein distances and barycenters on CPU. These can also be implemented on the GPU to further boost the efficiency, although it suffices to run on CPU for now, as evident from the timings below.

### S1.4 Timing information

The following timing benchmarks are done on 1 Nvidia V100 GPU. The time taken to average two MLPNET models for MNIST is $\approx 3$ seconds. For averaging VGG11 models on CIFAR10, it takes about $\approx 5$ seconds. While in case of RESNET18 on CIFAR10, it takes $\approx 7$ seconds. These numbers are for the activation-based alignment, and also include the time taken to compute the activations over the mini-batch of examples.

The weight-based alignment can be faster as it does not need to compute the activations. For instance, when weight-based alignment is employed to average two VGG11 models on CIFAR10, it takes $\approx 2.5$ seconds.

## S2 Ablation studies

### S2.1 Aggregation performance as training progresses

We compare the performance of averaged models at various points during the course of training the individual models (for the setting of MLPNet on MNIST). We notice that in the early stages of training, vanilla averaging performs even worse, which is not the case for OT averaging. The corresponding Figure S2 and Table S1 can be found in Section S2.1 of the Appendix. Overall, we see OT averaging outperforms vanilla averaging by a large margin, thus pointing towards the benefit of aligning the neurons via optimal transport.

Figure S2: Illustrates the performance of various aggregation methods as training proceeds, for (MNIST, MLPNET). The plots correspond to the results reported in Table S1. The activation-based alignment of the OT average (labelled as structure-aware accuracy in the figure) is used based on $m = 200$ samples.

### S2.2 Transport map for the output layer.

Since our algorithm runs until the output layer, we inspect the alignment computed for the last output layer. We find that the ratio of the trace to the sum for this last transport map is $\approx 1$, indicating accurate alignment as the ordering of output units is the same across models.

---

[S1] http://pot.readthedocs.io/en/stable/

| EPOCH | MODEL A | MODEL B | PREDICTION AVG. | VANILLA AVG. | OT AVG. |
|---|---|---|---|---|---|
| 01 | 92.03 | 92.40 | 92.50 | 47.39 | 87.10 |
| 02 | 94.39 | 94.43 | 94.79 | 52.28 | 91.72 |
| 05 | 96.83 | 96.58 | 96.93 | 58.96 | 95.30 |
| 07 | 97.36 | 97.34 | 97.48 | 68.76 | 95.26 |
| 10 | 97.72 | 97.75 | 97.88 | 73.84 | 95.92 |
| 15 | 97.91 | 97.97 | 98.11 | 73.55 | 95.60 |
| 20 | 98.11 | 98.04 | 98.13 | 73.91 | 95.31 |

Table S1: **Activation-based alignment (MNIST, MLPNet):** Comparison of performance when ensembled after different training epochs. The # samples used for activation-based alignment, $m = 50$. The corresponding plot for this table is illustrated in Figure S2.

## S2.3 Effect of mini-batch size needed for activation-based mode

Here, the individual models used are MLPNET's which have been trained for 10 epochs on MNIST. They differ only in their seeds and thus in the initialization of the parameters alone. We ensemble the final checkpoint of these models via OT averaging and the baseline methods.

| $M_A$ | $M_B$ | PREDICTION AVG. | VANILLA AVG. | $m$ | OT AVG. (SINKHORN) Accuracy (mean $\pm$ stdev) | $M_A$ ALIGNED |
|---|---|---|---|---|---|---|
| *(a) Activation-based Alignment* | | | | | | |
| 97.72 | 97.75 | 97.88 | 73.84 | 2 | $24.80 \pm 6.93$ | $20.08 \pm 2.42$ |
| | | | | 10 | $75.04 \pm 11.35$ | $88.18 \pm 8.45$ |
| | | | | 25 | $90.95 \pm 3.98$ | $95.36 \pm 0.96$ |
| | | | | 50 | $93.47 \pm 1.69$ | $96.04 \pm 0.59$ |
| | | | | 100 | $95.40 \pm 0.52$ | $\mathbf{97.05 \pm 0.17}$ |
| | | | | 200 | $\mathbf{95.78 \pm 0.52}$ | $97.01 \pm 0.16$ |
| *(b) Weight-based Alignment* | | | | | | |
| 97.72 | 97.75 | 97.88 | 73.84 | — | 95.66 | 96.32 |

Table S2: One-shot averaging for (MNIST, MLPNet) **with Sinkhorn and regularization** $= 0.05$: Results showing the performance (i.e., test classification accuracy (in %)) of the OT averaging in contrast to the baseline methods. The last column refers to the aligned model A which gets (vanilla) averaged with model B, giving rise to our OT averaged model. $m$ is the size of mini-batch over which activations are computed.

## S2.4 Effect of regularization

The results for activation-based alignment presented in the Table S2 above use the regularization constant $\lambda = 0.05$. Below, we also show the results with a higher regularization constant $\lambda = 0.1$. As expected, we find that using a lower value of regularization constant leads to better results in general, since it better approximates OT.

| $M_A$ | $M_B$ | PREDICTION | VANILLA | $m$ | OT AVG. Accuracy (mean $\pm$ stdev) | $M_A$ aligned |
|---|---|---|---|---|---|---|
| 97.72 | 97.75 | 97.88 | 73.84 | 2 | $25.05 \pm 7.22$ | $19.42 \pm 2.28$ |
| | | | | 10 | $72.86 \pm 11.93$ | $74.35 \pm 14.40$ |
| | | | | 25 | $89.49 \pm 5.21$ | $90.88 \pm 4.91$ |
| | | | | 50 | $92.88 \pm 2.03$ | $94.54 \pm 1.36$ |
| | | | | 100 | $95.14 \pm 0.49$ | $96.42 \pm 0.39$ |
| | | | | 200 | $\mathbf{95.70 \pm 0.54}$ | $\mathbf{96.63 \pm 0.23}$ |

Table S3: Activation-based alignment (MNIST, MLPNet) **with Sinkhorn and regularization** $= 0.1$: Results showing the performance (i.e., test classification accuracy ) of the averaged and aligned models of OT based averaging in contrast to vanilla averaging of weights as well as the prediction based ensembling. $m$ denotes the number of samples over which activations are computed, i.e., the mini-batch size.

## S2.5 Exact vs regularized variant

In Table S4, we contrast the results obtained when no regularization is used and exact optimal transport is considered. Since using the exact optimal transport is fast enough, we default to using it hereafter.

| $M_A$ | $M_B$ | PREDICTION AVG. | VANILLA AVG. | ALIGNMENT TYPE | OT AVG. | $M_A$ ALIGNED Accuracy (mean) |
|---|---|---|---|---|---|---|
| *Regularized OT (via Sinkhorn)* | | | | Activation | 95.78 | 97.01 |
| 97.72 | 97.75 | 97.78 | 73.84 | Weight | 95.66 | 96.32 |
| *Exact OT* | | | | Activation | 96.21 | 97.72 |
| 97.72 | 97.75 | 97.78 | 73.84 | Weight | 96.63 | 97.72 |

Table S4: **Exact vs Regularized OT:** Results showing the performance gain with exact OT for activation/weight based alignment. Here, regularization $\lambda = 0.05$.

## S2.6 Layer-wise Optimal Transport distances

Figure S3: Illustrates the layerwise Optimal Transport costs between the corresponding layers of two ResNet18 models trained from different initializations, when using activation-based alignment with mini-batch size $m = 200$.

A possible application of our model fusion approach can be for inspecting the similarity of representations at various layers across different neural networks. Thus, it could provide an alternative perspective for this problem of understanding the similarity of representations, besides the canonical correlation analysis (CCA) based methods used in the past [30]. Figure S3 gives an example of this for two ResNet18 models trained from different initializations. Here, we used activation-based alignment with mini-batch size $m = 200$. An extensive study, however, remains beyond the scope of this paper.

# S3 Detailed finetuning results

In Tables S5, S7, and S8, we report the results of finetuning (i.e. retraining) the averaged models for (MNIST, MLPNET) and (CIFAR10, VGG11). For comparison, we also show the performance of individual models when further finetuned in this setting. Although in general, **the individual model finetuning is not realistic**, since it is not known which one will lead to an improvement and this incurs # models $\times$ the finetuning cost.

## S3.1 Two model scenario

### S3.1.1 For MNIST + MLPNET

The finetuning is carried out for 60 epochs at the following set of constant learning rates $\{0.01, 0.002, 0.001, 0.00067, 0.0005\}$. Note that the original models were trained for 10 epochs at a learning rate of $0.01$. For OT average, we use the activation-based alignment with mini-batch size $m = 200$.

Table S5 shows the results for each method at their best respective finetuning runs.

| FINETUNING LR | MODEL A | MODEL B | VANILLA AVG. | OT AVG. (EXACT) |
|---|---|---|---|---|
| *Baseline Results* | | | | |
| — | 97.72 | 97.75 | 73.84 | 96.54 |
| *Results for the **best** finetuning run* (reported at the best checkpoint) | | | | |
| 0.01 | 98.21 | 98.13 | 98.23 | **98.35** |
| 0.002 | 98.13 | 98.03 | 98.13 | **98.21** |
| 0.001 | 98.09 | 98.03 | 97.98 | **98.14** |
| 0.00067 | **98.11** | 98.00 | 97.83 | 98.07 |
| 0.0005 | **98.09** | 98.01 | 97.70 | 98.05 |

Table S5: **Effect of finetuning the individual and averaged models for (MNIST, MLPNet):** Best finetuning runs have been reported for each method. Cells in orange highlight the best scores in each regime.

We also show in Table S6 the results when averaged across 5 finetuning runs for each of the finetuning LR, as the cost of finetuning here is not as prohibitive in comparison to finetuning VGG11 and ResNet18 models. We see that performance trend remains in accordance with the previous Table S5.

| FINETUNING LR | MODEL A | MODEL B | VANILLA AVG. | OT AVG. (EXACT) |
|---|---|---|---|---|
| *Baseline Results* | | | | |
| — | 97.72 | 97.75 | 73.84 | $96.21 \pm 0.36$ |
| ***Averaged** results across the finetuning runs* (reported at the best checkpoint) | | | | |
| 0.01 | $98.19 \pm 0.02$ | $98.11 \pm 0.02$ | $98.22 \pm 0.02$ | $\mathbf{98.28 \pm 0.05}$ |
| 0.002 | $98.13 \pm 0.01$ | $98.03 \pm 0.01$ | $98.13 \pm 0.01$ | $\mathbf{98.15 \pm 0.07}$ |
| 0.001 | $\mathbf{98.11 \pm 0.02}$ | $98.01 \pm 0.01$ | $97.99 \pm 0.01$ | $98.08 \pm 0.05$ |
| 0.00067 | $\mathbf{98.11 \pm 0.02}$ | $98.00 \pm 0.01$ | $97.83 \pm 0.02$ | $98.05 \pm 0.04$ |
| 0.0005 | $\mathbf{98.09 \pm 0.01}$ | $98.01 \pm 0.00$ | $97.68 \pm 0.01$ | $98.03 \pm 0.03$ |

Table S6: **Effect of finetuning the individual and averaged models for (MNIST, MLPNet):** Average of the results across 5 finetuning runs as well as their standard deviation are reported for each method. Cells in orange highlight the best scores in each regime.

### S3.1.2   For CIFAR10 + VGG11

As a recall, the original models were trained for 300 epochs at an initial learning rate of $0.05$, which was decayed by a factor of 2 after every 30 epochs. The finetuning is carried out for 100 epochs at the following set of initial learning rates $\{0.01, 0.05, 0.0033, 0.0025\}$. Also, similar to training, the learning rate is decayed in the finetuning process. Note that, here finetuning at the initial learning rate of $0.01$ causes model B to diverge and hence we skip the results for this setting.

For OT average, we use the weight-based alignment. Table S7 shows the best results for each method during their finetuning run.

| FINETUNING LR | MODEL A | MODEL B | VANILLA AVG. | OT AVG. (EXACT) |
|---|---|---|---|---|
| *Baseline Results* | | | | |
| — | 90.31 | 90.50 | 17.02 | 85.98 |
| *Results after finetuning* (reported scores are at best checkpoint) | | | | |
| 0.01 | 90.29 | 90.53 | 90.39 | **90.73** |
| 0.005 | 90.36 | 90.47 | 90.16 | **90.64** |
| 0.0033 | 90.28 | 90.39 | 90.13 | **90.39** |
| 0.0025 | 90.45 | **90.50** | 89.88 | 90.30 |

Table S7: **Effect of finetuning the individual and averaged models for (CIFAR10, VGG11):** Model A & Model B baseline accuracies correspond to best checkpoints when originally trained for 300 epochs. Cells in orange highlight the best scores in each regime.

### S3.1.3   For CIFAR10 + RESNET18

As a recall, the original models were trained for 300 epochs at an initial learning rate of $0.1$, which was decayed by a factor of 10 at the epochs $\{150, 250\}$. The finetuning is carried out for 120 epochs at the following set of initial learning rates $\{0.1, 0.04, 0.02\}$. For OT average, we use the activation-based alignment, with mini-batch size $m = 200$.

| FINETUNING LR | MODEL A | MODEL B | VANILLA AVG. | OT AVG. (EXACT) |
|---|---|---|---|---|
| *Baseline Results* | | | | |
| — | 93.11 | 93.20 | 18.49 | 67.46 |
| *Results after finetuning* (reported at the best checkpoint) | | | | |
| (a) *LR decay epochs* $= [20, 40, 60, 80, 100]$ | | | | |
| 0.1 | 93.51 | 93.43 | 93.29 | **93.78** |
| 0.04 | 93.35 | 93.34 | 93.28 | **93.35** |
| 0.02 | 93.28 | **93.28** | 93.09 | 92.97 |
| (b) *LR decay epochs* $= [40, 80]$ | | | | |
| 0.1 | 93.49 | 93.32 | 93.34 | **93.59** |
| 0.04 | 93.27 | 93.34 | **93.49** | 93.38 |
| 0.02 | 93.21 | **93.33** | 93.17 | 93.15 |

Table S8: **Effect of finetuning the individual and averaged models for (CIFAR10, RESNET18):** Model A and Model B baseline accuracies correspond to best checkpoints when originally trained for 300 epochs. Cells in orange highlight the best scores in each regime.

Table S8 shows the best results for each method during their finetuning run. The learning rate is decayed by a factor of 2 in the finetuning process as per two schedules: (a) after every 20 epochs, and (b) after every 40 epochs. These are indicated in the respective sections of the Table S8.

## S3.2 Multiple model scenario: CIFAR10

Now, we discuss in detail, the experiments performed for the multiple model setting on CIFAR10. Namely, when we have 4 and 6 VGG11 models, that have different initializations, but are trained identically on the entire data, as mentioned in Table S9.

| CIFAR10+ VGG11 | INDIVIDUAL MODELS | PREDICTION AVG. | VANILLA AVG. | OT AVG. | FINETUNING VANILLA | FINETUNING OT |
|---|---|---|---|---|---|---|
| Accuracy | [90.31, 90.50, 90.43, 90.51] | 91.77 | 10.00 | 73.31 | 12.40 | 90.91 |
| Efficiency | $1\times$ | $1\times$ | $4\times$ | $4\times$ | $4\times$ | $4\times$ |
| Accuracy | [90.31, 90.50, 90.43, 90.51, 90.49, 90.40] | 91.85 | 10.00 | 72.16 | 11.01 | 91.06 |
| Efficiency | $1\times$ | $1\times$ | $6\times$ | $6\times$ | $6\times$ | $6\times$ |

Table S9: Results of our OT average + finetuning based efficient alternative for ensembling in contrast to vanilla average + finetuning, for more than two input models (VGG11) with different initializations trained on CIFAR10.

We consider finetuning the averaged models, with many different optimization hyperparameters, however vanilla average fails to finetune or retrain. In particular, we finetune for 150 epochs with learning rate obtained by dividing the original learning rate (with which models were trained) by factors of $\{1, 2, 4, 8, 16\}$ (called 'initial decay'). Further, similar to learning rate schedule followed in the training, we try decaying the learning rate by a factor of $\{1.1, 1.5, 2.0\}$ after every 20 epochs. We also tried adjusting the interval after which the learning rate was decayed (like 40 epochs), but this was again to no avail in being able to finetune the vanilla average. So for simplicity, in the rest of discussion, we consider that the interval after which the learning rate gets decayed is 20 epochs.

Across all the settings OT average is able to successfully retrain, except when the learning rate is set to the original learning rate of $0.05$, with which models were trained (i.e., initial decay of 1). This is to be expected as the OT average without retraining itself already performs fairly well, and setting such a high learning rate is bound to cause this. In contrast, vanilla average fails to retrain at all, with the best accuracy of $12.40$ and $11.01$ for the case of 4 and 6 models, when the initial decay is 1, and the learning rate decay is $1.1$.

Finetuning from OT average results, in a significant improvement for numerous settings of the above hyperparameters, and below, we show the top 5 such settings in Table S10 for both 4 and 6 models. (For OT average, we use the activation-based alignment.)

| INITIAL DECAY FACTOR | SCHEDULED LR DECAY FACTOR | DECAY INTERVAL | FINETUNING VANILLA AVG. | FINETUNING OT AVG. |
|---|---|---|---|---|
| *(i) Number of models $= 4$* | | | | |
| 2 | 2.0 | 20 | 10.34 | 90.91 |
| 4 | 2.0 | 20 | 10.32 | 90.80 |
| 2 | 2.0 | 40 | 10.34 | 90.74 |
| 2 | 1.5 | 20 | 10.34 | 90.67 |
| 4 | 2.0 | 40 | 10.32 | 90.66 |
| *(ii) Number of models $= 6$* | | | | |
| 2 | 2.0 | 20 | 10.00 | 91.06 |
| 2 | 1.5 | 20 | 10.00 | 90.97 |
| 4 | 2.0 | 20 | 10.00 | 90.88 |
| 4 | 2.0 | 40 | 10.00 | 90.81 |
| 8 | 2.0 | 40 | 10.00 | 90.69 |

Table S10: Different finetuning settings which show how OT fusion can improve over the individual models after finetuning, while the vanilla average fails to do so. As a result, we obtain one single improved model that can be used as an efficient replacement for the ensemble.

# S4  Finetuning curves

| Name | Smoothed | Value | Step | Time | Relative |
|---|---|---|---|---|---|
| ● test_accuracy_percent/geometric_21 | 90.20 | 90.60 | 65.00 | Wed Sep 18, 02:47:59 | 38m 10s |
| ○ test_accuracy_percent/naive_averaging | 90.15 | 90.31 | 65.00 | Wed Sep 18, 03:46:35 | 38m 5s |

Figure S4: Illustrates the performance of OT averaging (referred to as geometric in the figure legend) and vanilla averaging during the process of retraining for CIFAR10 with VGG11.

| Name | Smoothed | Value | Step | Time | Relative |
|---|---|---|---|---|---|
| ● test_accuracy_percent/geometric_21 | 90.20 | 90.60 | 65.00 | Wed Sep 18, 02:47:59 | 38m 10s |
| ● test_accuracy_percent/model_0 | 90.22 | 90.31 | 65.00 | Wed Sep 18, 01:28:37 | 17m 34s |
| ○ test_accuracy_percent/model_1 | 90.11 | 90.17 | 65.00 | Wed Sep 18, 01:55:29 | 17m 29s |
| ○ test_accuracy_percent/naive_averaging | 90.15 | 90.31 | 65.00 | Wed Sep 18, 03:46:35 | 38m 5s |

Figure S5: Retraining with reference plots of individual models. Other than that same as above.

# S5 Skill Transfer: Additional Results

## S5.1 Remaining Data Split: $10\%$

(a) Special digit 4, same init avg

(b) Special 6, same init avg

(c) Special digit 6, different init avg

Figure S6: **Skill Transfer performance**: Comparison results of OT based model fusion (OT avg) with vanilla averaging for different $w_B$. Each point for OT avg. curve (magenta colored) is obtained by activation-based alignment with a batch size $m = 400$, and we plot the mean performance over 5 seeds along with the error bars, which show the corresponding standard deviation. Here the remaining data besides the special digit, is split as $90\%$ for model B and the other $10\%$ for model A.

## S5.2 Remaining Data Split: 5%

(a) Special digit 4, same init avg

(b) Special digit 4, different init avg

(c) Special digit 6, same init avg

(d) Special digit 6, different init avg

Figure S7: **Skill Transfer performance**: Comparison results of OT based model fusion (OT avg) with vanilla averaging for different $w_B$. Each point for OT avg. curve (magenta colored) is obtained by activation-based alignment with a batch size $m = 400$, and we plot the mean performance over 5 seeds along with the error bars, which show the corresponding standard deviation. Here the remaining data besides the special digit, is split as 95% for model B and the other 5% for model A.

## S5.3 Scenarios without specialized labels

Even if we don't exclude a digit and just alter the fraction of data between A and B, results are similar. E.g., take MLPNETS A and B with *same* initialization (*to help vanilla averaging*), but A has $30\%$ and B has $70\%$ of the data. This results in (global) test accuracy $\%$ of 94.2 and 95.0 for A and B resp. OT fusion is better than vanilla averaging when combining A and B for all proportions, with best results as, OT: mean **95.3** (stdev=0.1), vanilla avg: 95.1 at proportions $0.1, 0.9$ respectively. Ensembling is better than both (95.5), but requires 2x more memory and inference time.

Likewise, for other data splits (such as $10\%$ vs $90\%$, $50\%$ vs $50\%$, etc), OT fusion outperforms the individual models as well as vanilla averaging. For, further settings, also see Section S7.

## S6 Results for one-shot skill-transfer under size constraints

Here, we present results for one-shot skill-transfer when the two models are of unequal sizes. More concretely, as an example, we consider that the hidden layers of the generalist model B are twice as wide as that of the specialist model A. Figure S8 illustrates the results for OT-based model fusion (OT average) in such a setting. Note that, here vanilla averaging can not be applied as the models are of different sizes. To the best of our knowledge, we are unaware of any other method that can allows for such one-shot skill transfer (i.e., fuse the given different size models into a single model in one-shot).

(a) Special digit 4, data split% = 10

(b) Special digit 4, data split% = 5

(c) Special digit 6, data split% = 10

(d) Special digit 6, data split% = 5

Figure S8: **Skill Transfer performance for different sized models**: Results of OT-based model fusion (OT avg) for different $w_B$. Unlike the results in the previous section, vanilla averaging is not possible here as the models are of unequal sizes. 'Width-Ratio 0.5' in the figure title denotes the ratio of the hidden layers sizes of model A and B. Each point for OT avg. curve (magenta colored) is obtained by activation-based alignment with a batch size $m = 400$, and we plot the mean performance over 5 seeds along with the error bars, which show the corresponding standard deviation. The data split % indicates the amount of remaining data besides the special digit which is present with model A. Model B contains 100 - data split% of this remaining data.

Rest of the technical details are identical as in the setup of Sections 5.1 in the main text and S5 in the supplementary.

## S7  Multi-model one-shot skill transfer

To recap, here we take four MLPNET models: A, B, C and D, with the same initialization and assume that A again possessing the knowledge of a special digit (say, 4). Consider that the rest of the data is divided as $10\%, 30\%, 50\%, 10\%$.

Now they are trained in a similar setting for 10 epochs, by the end of which these models obtain (global) test accuracies of $87.7\%, 86.5\%, 87.0\%, 83.5\%$ respectively. Since A is the only model which has seen the special digit '4', we assign it a larger proportion in the final fused model. In particular, we consider fusing the models in proportions of $0.7, 0.1, 0.2, 0.1$ respectively *(later normalized to sum to 1)*. Then, ensembling the predictions yields $95.0\%$ while vanilla averaging obtains $80.6\%$. In contrast, OT averaging results in **93.6%** test accuracy ($\approx 6\%$ gain over the best individual model), while being $4\times$ more efficient than ensembling.

This is also robust to many other proportions in which the models are combined. For example, decreasing the weight of model A so that the proportions are $0.6, 0.1, 0.2, 0.1$, gives: Prediction ensembling $95.03\%$, vanilla average $78.44\%$, OT average **92.72%**. Or increasing the proportion of B and D, i.e., let the proportions be instead $0.7, 0.15, 0.2, 0.15$. The results for such a case are as follows, Prediction ensembling $94.91\%$, vanilla average $76.14\%$, OT average $91.67\%$. Take another example, say we increase the proportion of model C now, so as to have the proportions $0.7, 0.1, 0.3, 0.1$. In this case, we get Prediction ensembling $95.15\%$, vanilla average $77.93\%$, OT average **92.21**%. We can go on for many other examples, but the results remain similar.

Overall, we find that OT average leads to a significant across all these examples, and outperforms vanilla average by a large margin. In comparison to prediction ensembling, it is slightly worse in terms of accuracy, but it enjoys $4\times$ efficiency, with respect to future usage and maintenance.

## S8  Post-processing for structured pruning

**CIFAR10.**   In this section, we present the detailed results for using OT fusion as a post-processing tool for structured pruning. We show the benefit gained by OT fusion when separately pruning all layers of VGG11, as well as pruning them all together. This is illustrated for the three cases: (a) when filters with smallest $\ell_1$ norms are removed, (b) when filters with smallest $\ell_2$ norms are removed, and (c) when filters are removed randomly, in Figures S9, S10, and S11 respectively.

**CIFAR100.**   Also in Figures S12, we show the results of a similar experiment when pruning a VGG11 model trained on CIFAR100. Here as well, OT fusion leads to a performance boost when used as for post-processing. For simplicity, we only include the results with $\ell_1$-pruner.

(a) conv_1

(b) conv_2

(c) conv_3

(d) conv_4

(e) conv_5

(f) conv_6

(g) conv_7

(h) conv_8

(i) all

Figure S9: Post-processing for structured pruning **with $\ell_1$ norm**, all figures: Fusing the initial dense VGG11 model into the pruned model helps test accuracy of the pruned model on **CIFAR10**.

(a) conv_1

(b) conv_2

(c) conv_3

(d) conv_4

(e) conv_5

(f) conv_6

(g) conv_7

(h) conv_8

(i) all

Figure S10: Post-processing for structured pruning **with $\ell_2$ norm**, all figures: Fusing the initial dense VGG11 model into the pruned model helps test accuracy of the pruned model on **CIFAR10**.

(a) conv_1

(b) conv_2

(c) conv_3

(d) conv_4

(e) conv_5

(f) conv_6

(g) conv_7

(h) conv_8

(i) all

Figure S11: Post-processing for structured pruning **with random**, all figures: Fusing the initial dense VGG11 model into the pruned model helps test accuracy of the pruned model on **CIFAR10**. Results are averaged over 3 seeds.

(a) conv_1

(b) conv_2

(c) conv_3

(d) conv_4

(e) conv_5

(f) conv_6

(g) conv_7

(h) conv_8

(i) all

Figure S12: Post-processing for structured pruning **with $\ell_1$ norm**, all figures: Fusing the initial dense VGG11 model into the pruned model helps test accuracy of the pruned model on **CIFAR100**.

## S9  Additional discussion on the update rule in the algorithm

### S9.1  Barycentric projection

The original formulation by [1] considers finding a mapping $f : \boldsymbol{S}_A \mapsto \boldsymbol{S}_B$, which maps points in the support of $\mu_A$ to points in the support of $\mu_B$. However, under this formulation, the optimal transport problem is not always feasible and Kantorovich [2] relaxed this by instead considering the optimization over the set of coupling matrices $\boldsymbol{T}$ (i.e., doubly stochastic matrices). Hence, a simple way to obtain the optimal map $f$ from this coupling/transportation matrix $\boldsymbol{T}$ is $f(\boldsymbol{S}_A^i) = \boldsymbol{Z}_i$, where $\boldsymbol{Z} = \boldsymbol{S}_B \boldsymbol{T}^\top \operatorname{diag}\left(\alpha^{-1}\right)$, c.f. Ferradans et al. [31]. Essentially, this maps each point in support of A to a weighted average of the points in support of B. This is referred to as the barycentric projection.

For our algorithm in the weight-based alignment, we basically need the opposite map $f' : \boldsymbol{S}_B \mapsto \boldsymbol{S}_A$ to get the weights for the model A aligned with respect to B. This can be done by simply exchanging the above supports and transposing the matrix $\boldsymbol{T}$. Thus we have, $f'(\boldsymbol{S}_B^i) = \boldsymbol{Z}_i'$, where $\boldsymbol{Z}' = \boldsymbol{S}_A \boldsymbol{T} \operatorname{diag}\left(\beta^{-1}\right)$. In other words, we represent each point in the support of B with a weighted average of points in support of A.

Lastly, the supports in this weight-based alignment are defined by the corresponding weight matrices of the layers. Thus, implying the update in Eq. (1). Note that, when the underlying ground cost used is the squared Euclidean distance, this barycentric mapping is known to be optimal [32].

### S9.2  Free-support barycenters

Next, we discuss the setup and a part of the derivation of free-support barycenters proposed in [4].

**Problem formulation.**  Assume that the ground metric $D_{\mathcal{S}}$ is the Euclidean distance and $p = 2$. Consider the supports $\boldsymbol{S}_A$ and $\boldsymbol{S}_B$ as family of $n_A$ and $n_B$ points respectively in $\mathbb{R}^d$. Therefore represent them by a matrix in $\mathbb{R}^{d \times n_A}$ and $\mathbb{R}^{d \times n_B}$ respectively. If we use the notation, $\mathbf{s_A} \overset{\text{def}}{=} \operatorname{diag}\left(\boldsymbol{S}_A^\top \boldsymbol{S}_A\right)$ and $\mathbf{s_B} \overset{\text{def}}{=} \operatorname{diag}\left(\boldsymbol{S}_B^\top \boldsymbol{S}_B\right)$, then we can write the pairwise squared-Euclidean distances as follows:

$$\boldsymbol{C}_{AB} = \mathbf{s_A} \mathbf{1}_{n_B}^\top + \mathbf{1}_{n_A} \mathbf{s_B}^\top - 2\boldsymbol{S}_A^\top \boldsymbol{S}_B \in \mathbb{R}^{n_A \times n_B}. \tag{5}$$

Now the optimal transport objective mentioned in Section 3 can be written in a more compact form in the equation (6) by using the matrix inner product notation. As a recall, $\langle \boldsymbol{U}, \boldsymbol{V} \rangle = \operatorname{tr}(\boldsymbol{U}^\top \boldsymbol{V})$, so we have:

$$
\begin{aligned}
\langle \boldsymbol{T}, \boldsymbol{C}_{AB} \rangle &= \left\langle \boldsymbol{T}, \mathbf{s_A} \mathbf{1}_d^\top + \mathbf{1}_d^\top \mathbf{s_B} - 2\boldsymbol{S}_A^\top \boldsymbol{S}_B \right\rangle \\
&= \operatorname{tr}(\boldsymbol{T}^\top \mathbf{s_A} \mathbf{1}_d^\top) + \operatorname{tr}(\boldsymbol{T}^\top \mathbf{1}_d^\top \mathbf{s_B}) - 2 \left\langle \boldsymbol{T}, \boldsymbol{S}_A^\top \boldsymbol{S}_B \right\rangle \\
&= \mathbf{s_A}^\top \alpha + \mathbf{s_B}^\top \beta - 2 \left\langle T, \boldsymbol{S}_A^\top \boldsymbol{S}_B \right\rangle.
\end{aligned}
\tag{6}
$$

Suppose we are given the transport map $\boldsymbol{T}^\star$ which is optimal for the above Eq (6), but we do not know the support $\boldsymbol{S}_B$. One way to compute it is by minimizing the above with respect to $\boldsymbol{S}_B$. Hence, let's discard the constant terms in $\mathbf{s_A}$ and $\alpha$. Recall $\mu_A = (\alpha, \boldsymbol{S}_A)$ and $\mu_B = (\beta, \boldsymbol{S}_B)$, so we have that minimizing $\operatorname{OT}\left(\mu_A, \mu_B, \boldsymbol{C}_{AB}\right)$ with respect to the locations $\boldsymbol{S}_B$ is same as solving

$$\min_{\boldsymbol{S}_B \in \mathbb{R}^{d \times n_B}} \mathbf{s_B}^\top \beta - \left\langle \boldsymbol{T}^\star, \boldsymbol{S}_A^\top \boldsymbol{S}_B \right\rangle \tag{7}$$

**Quadratic Approximation.**  Cuturi and Doucet [4] show that the above minimization problem in Eq. (7) is non-convex in the locations $\boldsymbol{S}_B$, the proof of which can be found in their work. Therefore, they resort to a local quadratic approximation mentioned in equation (8), minimizing which yields the Newton update in equation (9).

$$
\begin{aligned}
\mathbf{s_B}^\top \beta - \left\langle \boldsymbol{T}^\star, \boldsymbol{S}_A^\top \boldsymbol{S}_B \right\rangle = & \left\| \boldsymbol{S}_B \operatorname{diag}\left(\beta^{1/2}\right) - \boldsymbol{S}_A \boldsymbol{T}^\star \operatorname{diag}\left(\beta^{-1/2}\right) \right\|^2 \\
& - \left\| \boldsymbol{S}_A \boldsymbol{T}^\star \operatorname{diag}\left(\beta^{-1/2}\right) \right\|^2
\end{aligned}
\tag{8}
$$

$$\boldsymbol{S}_B \leftarrow \boldsymbol{S}_A \boldsymbol{T}^\star \operatorname{diag}\left(\beta^{-1}\right) \qquad (9)$$

This can be interpreted as follows as follows: the matrix $\boldsymbol{T}^\star \operatorname{diag}\left(\beta^{-1}\right)$ has $n_B$ columns in the simplex $\Sigma_{n_A}$ and thus post-multiplying $\boldsymbol{S}_A$ with this matrix means that we are performing convex combinations of the points in $\boldsymbol{S}_A$ with weights defined by the optimal transport map $\boldsymbol{T}^\star$.

**Relation to Model Fusion.** Let's come back to our algorithm where we use the weight-based alignment. Here, the locations (or the supports) are defined by the corresponding weight matrices of the layers. This bears resemblance to the update in Eq. (1) in Section 4, where the equivalent of the unknown $\boldsymbol{S}_B$ are the weights of A aligned with respect to B.

## S10  Connection to the mean-field limit

| HIDDEN LAYERS | % TEST ACCURACY | | | | | % RELATIVE GAP | |
|---|---|---|---|---|---|---|---|
| | MODEL A | MODEL B | PREDICTION AVG. | VANILLA AVG. | OT AVG. | VANILLA AVG. | OT AVG. |
| $[40, 20, 10]$ | 96.69 | 96.91 | 97.50 | 34.82 | 82.91 | 64.07 | **14.44** |
| $[200, 100, 50]$ | 98.00 | 97.97 | 98.16 | 47.30 | 93.93 | 51.73 | **4.16** |
| $[400, 200, 100]$ | 98.13 | 98.09 | 98.21 | 73.51 | 96.70 | 25.09 | **1.45** |
| $[1000, 500, 250]$ | 98.08 | 98.21 | 98.20 | 78.21 | 97.35 | 20.36 | **0.87** |
| $[2000, 1000, 500]$ | 98.26 | 98.16 | 98.21 | 85.71 | 97.41 | 12.77 | **0.86** |

Table S11: Relative gap of OT avg. wrt the best individual model as the width of the hidden layers increases.

*Effect of layer width:* Table S11 illustrates that as the width of networks increases, the gap in performance of one-shot OT averaging compared to the best individual network decreases. This also suggests a very interesting potential connection with the mean-field limit for neural networks [33].

Here, the authors show that as the size of the hidden layer goes to infinity, doing gradient descent on the network weights is equivalent to considering a probability density over the neurons in a layer, which evolves with Wasserstein gradient flow. Then given two neural networks evolving under the dynamics of Wasserstein gradient flow, fusing them into one network by Wasserstein barycenter would be a natural consideration.

We empirically show that this limit is roughly achieved in practice when the width $\approx 1000$. In particular, Table S11 illustrates that as the width of networks increases, the gap in performance of one-shot averaging (with respect to the best individual network) on MNIST decreases. As a consequence, this further implies that in the setting of finite hidden layer sizes, it would help to choose the $\alpha$ and $\beta$ in a better way than just uniform. We aim to study this aspect in a future work.

## S11 Teacher-Student Fusion

We present the results for a setting where we have trained teacher and student networks, and we would like to combine the knowledge of large teacher network into the smaller student network. This is essentially reverse of the client-server setting described in Section 5.2. We consider that all the hidden layers of the teacher model $M_A$, are a constant $\rho\times$ wider than all the hidden layers of student model $M_B$. We experiment with two instances of this (a) on MNIST + MLPNET, with $\rho \in \{2, 10\}$ and (b) on CIFAR10 + VGG11, with $\rho \in \{2, 8\}$, and the results are presented in the Table S12. This leads to the mentioned model sizes (# of parameters) for both these models. Our OT average uses activation-based alignment for both the settings described in the Table S12.

Vanilla averaging can not be used due to different sizes of the networks. So, as a first baseline, we consider the performance of finetuning the model $M_B$. We observe that across all the settings, OT avg. + finetuning outperforms this baseline as well as the original model $M_B$, resulting in the desired knowledge transfer from the teacher network.

| DATASET + MODEL | # params $(M_A, M_B)$ | $M_A$ | $M_B$ | OT AVG. | FINETUNING $M_B$ | FINETUNING OT AVG. |
|---|---|---|---|---|---|---|
| MNIST + MLPNET | (414 K, 182 K) | 98.11 | 97.84 | 95.67 | 98.06 | **98.22** |
| | (414 K, 32 K) | 98.11 | 97.08 | 96.50 | 97.31 | **97.42** |
| CIFAR10 + VGG11 | (118 M, 32 M) | 91.22 | 90.66 | 86.73 | 90.67 | **90.89** |
| | (118 M, 3 M ) | 91.22 | 89.38 | 88.40 | 89.64 | **89.85** |

Table S12: *Compressing $M_A$ to smaller models.* The finetuning results of each method are at their best scores across different finetuning hyperparameters (like, learning rate schedules). OT avg. has the same number of parameters as $M_B$.

| *Dataset + Model* | # Student params | Finetune type | LR | Epochs | Enabled | LR Decay Factor | LR Decay Epochs |
|---|---|---|---|---|---|---|---|
| MNIST + MLPNET | 182 K | $M_B$ | 0.01 | 60 | ✗ | — | — |
| | 32 K | | 0.002 | | | | |
| | 182 K | OT Avg. | 0.01 | | | | |
| | 32 K | | 0.001 | | | | |
| CIFAR10 + VGG11 | 32 M | $M_B$ | 0.01 | 120 | ✓ | 2.0 | $[20, 40, 60, 80, 100]$ |
| | 3 M | | | | | 1.5 | $[10, 30, 50, 70, 90, 110]$ |
| | 32 M | OT Avg. | 0.01 | | | 2.0 | $[20, 40, 60, 80, 100]$ |
| | 3 M | | | | | | |

Table S13: Hyper-parameters corresponding to the results for model compression presented in Tables S12. LR denotes the learning rate. For the MNIST + MLPNET setting a constant learning rate was employed, similar to its training procedure. While for the CIFAR10 + VGG11 setting, the learning rate schedule was also tuned, keeping in accordance with its training procedure as well. The # params column indicates the size of the resultant compressed (smaller) model.

We show that even if the smaller model $M_B$ were to be finetuned with many different hyper-parameters, it does not outperform the performance gained by finetuning the OT average.

For MNIST + MLPNET, we did a sweep for the following set of finetuning learning rates $\{0.01, 0.002, 0.001, 0.00067, 0.0005\}$. These correspond to scaling the training learning rate by a factor $\{1, 5, 10, 15, 20\}$ respectively. Both OT average and the model $M_B$ were finetuned for 60 epochs using these choices as a constant learning rate.

Next, for CIFAR10 + VGG11, we additionally sweep for the hyper-parameters associated with the learning rate (LR) schedule during finetuning. Since unlike the MNIST case, the original models here used a decaying learning rate schedule. The sweep was carried out for the following set of values: LR decay factor = $\{1.2, 1.5, 2\}$, LR decay epochs =

$\{[20, 40, 60, 80, 100], [10, 30, 50, 70, 90, 110], [30, 60, 90]\}$. The learning rate (LR) itself was picked from $\{0.01, 0.005, 0.0033, 0.0025\}$ corresponding to scaling the training learning rate by a factor of $\{5, 10, 15, 20\}$ respectively, as done in Section S3.1.2 before. Table S13 thus indicates the hyper-parameter choice corresponding to the best results presented in Table S12.

From sweeping on the width-ratio $\rho = 8$ for CIFAR10+ VGG11 (i.e., when the smaller model has 3M params), we found that the learning rates $\{0.01, 0.005\}$ produced the best results for both the finetuning type/methods (namely, OT average and model $M_B$). Thus, while sweeping on the width-ratio $\rho = 2$ for CIFAR10+ VGG11 (i.e., when the smaller model has 32M params), we reduce the hyper-parameter space by restricting the learning rate from this set $\{0.01, 0.005\}$, while still sweeping the other sets of hyper-param values, in order to save on resource costs.

Besides these best results for each method, we find that even under the same hyper-parameter configuration, finetuning OT average leads to a better performance than finetuning the smaller model $M_B$ across multiple runs, for majority of the hyper-parameter settings. Overall, we conclude that the OT average and then finetuning successfully allows to transfer the performance from a bigger model into a smaller one.

## S12 Results for distillation

In this section, we present the results in relation to distilling the knowledge of a bigger model $M_A$ into a smaller model $M_B$. Here, we consider that one already has a trained version of both the models and we are interested in boosting the performance of the smaller model.

We discuss two ways of approaching this problem. One is to consider the OT average of the individual models and then finetune. The other option is to use distillation [8], where we augment the loss during finetuning with a term that essentially encourages the student's (smaller model) logit distribution (smoothed) to be close that of the teacher's (bigger model) logit distribution. The smoothing is done by raising the temperature $(T)$ in the final softmax. This loss term from distillation gets is weighted by a factor of $\gamma$ and a factor of $1 - \gamma$ is given to the usual finetuning loss.

The main drawback of distillation is that it requires searching for the optimal values of these hyper-parameters: temperature $T$ and loss-weighting factor $\gamma$. As evident from the Tables S16 and S17, even for MNIST+ MLPNET, depending on the size of the smaller model, the optimal hyper-parameter values can be quite different. This can be prohibitive when dealing with larger models or datasets.

Nevertheless, we compare the performance of both these approaches in Tables S14 and S15, for the setting of MNIST+ MLPNET with width-ratio $\rho = 2, 10$ (the ratio of hidden layers sizes of the larger to the smaller model) respectively. For distillation, we consider three possible initializations for the smaller (student) model: random, model $M_B$, and OT average of models $M_A$, $M_B$.

| $M_A$ | $M_B$ | PREDICTION AVG. | OT AVG. | FINETUNING $M_B$ | OT AVG. | DISTILLATION RANDOM | $M_B$ | OT AVG. |
|---|---|---|---|---|---|---|---|---|
| 98.11 | 97.84 | 98.10 | 95.49 | 98.04 | 98.19 | 98.18 | 98.22 | **98.30** |
| | | Mean across distillation temperatures | | | | 98.13 | 98.17 | **98.26** |

Table S14: *Compressing the bigger teacher model $M_A$ to half its size ($\rho = 2$).* The distillation scores for each student network initialization are taken for its best hyperparameter values. Both finetuning and distillation were run for 60 epochs using SGD with the same hyperparameters. Each entry has been averaged across 4 seeds.

| $M_A$ | $M_B$ | PREDICTION AVG. | OT AVG. | FINETUNING $M_B$ | OT AVG. | DISTILLATION RANDOM | $M_B$ | OT AVG. |
|---|---|---|---|---|---|---|---|---|
| 98.11 | 97.08 | 98.13 | 96.50 | 97.19 | 97.35 | 97.39 | 97.67 | **97.68** |
| | | Mean across distillation temperatures | | | | 97.21 | 97.55 | **97.59** |

Table S15: *Compressing the bigger teacher model $M_A$ to one-tenth of its size ($\rho = 10$).* The student model for distillation is initialized in 3 possible ways: random, OT avg., and model $M_B$. In the first row, the distillation scores are taken at its best hyperparameter values. Both finetuning and distillation were run for 60 epochs. Each entry in the table has been averaged across four seeds.

The choice of distillation hyper-parameters tried was: temperature $T = \{20, 10, 8, 4, 1\}$ and loss-weighting factor $\gamma = \{0.05, 0.1, 0.5, 0.7, 0.95, 0.99\}$. Tables S14 and S15 report the best scores across these hyper-parameter choices. Also, each each of the reported scores in the tables have been averaged across 4 seeds. The optimization parameters are same for both finetuning and distillation to ensure fair comparison. Namely, the learning rate $= 0.01$ and momentum $= 0.5$, and both were optimized with SGD for 60 epochs.

In terms of the results, we observe that initializing with OT average does the best in comparison to random and model $M_B$-based initializations. Further, we see that when the results for distillation with the other initializations are averaged across the distillation temperatures, the gain in performance pales in comparison to simply OT average and finetuning (c.f. Table S14).

| TEMPERATURE | DISTILLATION INITIALIZATIONS | | |
|:---:|:---:|:---:|:---:|
| $T$ | RANDOM ($\gamma$) | $M_B$ ($\gamma$) | OT AVG. ($\gamma$) |
| 20 | 98.13 (0.05) | 98.20 (0.10) | **98.26** (0.05) |
| 10 | 98.15 (0.05) | 98.19 (0.05) | **98.28** (0.05) |
| 8 | 98.18 (0.05) | 98.22 (0.05) | **98.28** (0.05) |
| 4 | 98.11 (0.10) | 98.21 (0.10) | **98.30** (0.05) |
| 1 | 98.06 (0.05) | 98.04 (0.05) | **98.17** (0.05) |
| *Mean* | 98.13 | 98.17 | **98.26** |

Table S16: *Distillation results for the setting of $\rho = 2$:* Best results for various distillation initializations are shown for all the tried temperatures ($T$) values. The corresponding choice of the loss-weighing factor ($\gamma$) for these best scores is indicated next to them in brackets. Each of the scores has been averaged over four seeds. The cell in orange indicates the top result across all hyper-parameter settings and methods.

| TEMPERATURE | DISTILLATION INITIALIZATIONS | | |
|:---:|:---:|:---:|:---:|
| $T$ | RANDOM ($\gamma$) | $M_B$ ($\gamma$) | OT AVG. ($\gamma$) |
| 20 | 97.25 (0.50) | 97.61 (0.70) | **97.68** (0.70) |
| 10 | 97.32 (0.70) | **97.67** (0.70) | 97.65 (0.70) |
| 8 | 97.38 (0.50) | **97.67** (0.70) | 97.65 (0.70) |
| 4 | 97.39 (0.70) | 97.53 (0.70) | **97.57** (0.99) |
| 1 | 96.73 (0.05) | 97.28 (0.95) | **97.40** (0.95) |
| *Mean* | 97.21 | 97.55 | **97.59** |

Table S17: *Distillation results for the setting of $\rho = 10$:* Best results for various distillation initializations are shown for all the tried temperatures ($T$) values. The corresponding choice of the loss-weighing factor ($\gamma$) for these best scores is indicated next to them in brackets. Each of the scores has been averaged over four seeds. The cell in orange indicates the top result across all hyper-parameter settings and methods.

Lastly, in Tables S16 and S17, we show the detailed results for each of the distillation initializations for each of the temperature values tried. These correspond respectively to the summarized results presented in Tables S14 and S15. We observe that distillation from OT average performs the best for most of the hyper-parameter settings, as well as in terms of the overall top performance for both the width-ratio $\rho$ settings.

To conclude, when distillation is out of the question due to resource constraints, OT average + finetuning can go a long way. While in cases where distillation is permissible, it can be advantageous to initialize with OT average when fusing a big model into a smaller one.