[Reviews · NeurIPS 2020]

Review 1

Summary and Contributions: The main contribution of this work is to introduce a layer-wise approach to fusion neurons and weights of neural networks. The key idea here is to consider the barycenter based fusion of networks.

Strengths: The main strength of the work is demonstrating improved efficacy for model fusion over typical approaches like averaging.

Weaknesses: See Additional feedback.

Correctness: The claims and method seem appropriate.

Clarity: The paper is well written.

Relation to Prior Work: The prior work is sufficiently discussed.

Reproducibility: Yes

Additional Feedback: I have some questions regarding the continual fusion of N>2 models and catastrophic forgetting. Although for the few models fused in this work I am wondering what the net effect would be by continually fusing models in this way. The reasoning for this inquiry is that barycentric fusing of multiple empirical distributions (in this work the networks are in a sense simply the empirical distribution) will ultimately "blur" the distributions. Similarly so, fusing multiple models I can see has overall "losing" the representational capacity with respect to the individual tasks each individual network has been learned on - unless there is some fundamental similarity between the underlying tasks (in which case no loss or even improvement may be observed). Can you see this as being the case and can you provide an intuition for your understanding - perhaps from a regularization perspective. ================== There is definitely a scope for further development of this work from which this work seems an acceptable and interesting baseline. The additional responses and discussion in the rebuttal provide extra clarity and so I increase my score to Accept. A thought comes to mind now that eluded me before: I would have actually loved to see how this performs on a very simple model rather than a complex one. Furthering this comment, I believe it would be illuminating for the authors to discuss their work (in the future of course) how this method "scales" to non-deep learning algorithms. Given the generality of the title, in this light, it seems somewhat inappropriate for such a focus on neural networks.


Review 2

Summary and Contributions: This paper proposes a layer-wise fusion algorithm for neural networks based on optimal transport of the parameters in each layer. For various applications, the proposed algorithm shows superior performance over the vanilla averaging.

Strengths: It is interesting to fuse several model parameters into a single model with only model parameters and has multiple applications such as federated learning. For various settings, the proposed fusion algorithm showed superior performance over the vanilla baseline. The potential to federated or decentralized learning seems to be an interesting point.

Weaknesses: The paper is not well-written, e.g., the algorithm part is not clearly organized and addressed. The only comparing methods are 'prediction ensembling' and 'vanilla averaging' across all experiments, which is not convincing and sufficient. For example, in pruning experiments, there are some published structured pruning methods [25-27]. Only very small datasets (MNIST, CIFAR10) are used in experiments. It is not clear if the network and model will overfit on such small-scale dataset.

Correctness: The claim is correct. The methodology is correct.

Clarity: No.

Relation to Prior Work: Yes. In related work, the authors addressed the relations and differences to previous.

Reproducibility: Yes

Additional Feedback: In my understanding, "acts" needs to make use of additional unlabled examples and forward for each of the K individual models. Under this setting, what is the advantages compared to the "prediction average" methods? Some typos: In line 126, "For e.g." In line 231, "e.g." lack of commas In line 250, "Thus, "


Review 3

Summary and Contributions: The paper proposes to use the formulation of Optimal Transport (OT) to align the channels/neurons in two/multiple different models, and then do a weight fusion by averaging. The use cases of such weight fusion is beneficial in cases like special/general multi-tasking, pruning, federated learning, ensembling. In each case, the experiment shows the OT fusion outperforms vanilla averaging of weights.

Strengths: Using Optimal Transport problem to match the order of channels/neurons is a intuitive application of an traditional algorithm to deep learning, and is shown to outperform vanilla averaging where we ignore order. The paper lists lots of use cases such as special and general model fusion, federated learning, pruning, ensembling. The appendix contains lots of detailed experiments and results, which helps interested readers to learn more.

Weaknesses: 1. The special and general models A and B which focus on 1 class and 9 classes in MNIST respectively seems a bit artificial to me. The other constraints introduced in the paper, such as no fine-tuning allowed, no joint training allowed (due to data privacy), are also a bit strange, as these approaches are widely used in common scenarios like pruning or multi-task learning. The method’s usefulness seems to only exist under these strict and sometimes artificial assumptions. Under the most straightforward application (ensembling), the method does not bring improvement without fine-tuning, and even with fine-tuning, the improvement over vanilla averaging is very marginal and I would consider them to be within the error bar of CIFAR-10 classification (0.3%). The paper could benefit from running these experiments with multiple seeds and report mean and stds. 2. Also, in my opinion, "Vanilla averaging” of weights does not form a strong baseline. Normally people don’t average the weights of two identical architectures element wisely, which is unlikely to produce meaningful performance, as also shown in the paper. The reason, as the author tried to address using OT, is that all positions in channel dimension in a convolution or linear layer are equivalent, thus averaging channel 1 and 2 from model A and B respectively makes no more sense than swapping them. The main baseline of the work should be vanilla ensembling, which the method only outperforms with fine-tuning, though. Vanilla averaging could be served as an illustration that the method works, but itself is not competitive. Even under the constraint that we want only one model and no training examples are given, there could possibly be more competent baselines. 3. In the case of pruning, I would intuitively imagine the channels get aligned with the large model correspond to the channels that survived the pruning, which are essentially the same channels in the small model. To what degree this is true? If this is largely true, why would a model benefit from fusion with (almost) itself? If not, why? 4. In figure S9(i), the caption says “all”, which I suppose should indicate all layers are pruned together, but the legends says “conv_9”. Which one is the case? In addition, I wonder whether the difference between vanilla and OT fusion still exists when fine-tuning is enabled, as is often the case in pruning. 5. It seems the paper did not specify the reason to use weight-based or activation-based alignment in each of the experiment. =======================Post Rebuttal=========================== The rebuttal address many of my concerns, e.g., about the application cases and the prior practice of averaging, and my opinion changes towards acceptance.

Correctness: Yes

Clarity: Largely. The math part is a bit abstract.

Relation to Prior Work: Yes

Reproducibility: Yes

Additional Feedback:


Review 4

Summary and Contributions: Authors propose a to fuse neural net models by aligning weights or activations using the optimal transport (using the wassertian barycenter) instead of vanilla averaging of corresponding weights. This approach outperforms vanilla averaging by a big marging and can be used as an alternative to model ensembling in constraint settings. It's is suitable for federated learning and does not required same size layers.

Strengths: The propose approach seems to work very well on tested datasets and have the potential of great impact among a large set of applications. - The comparison with ensembling methods is very appealing as well as the application to structured pruning.

Weaknesses: - Authors just compare against vanilla averaging of weights, but there are similar proposed approaches that are important to consider. I would really like to see a performance comparison with [1] for example. - The datasets and models where the idea was tested are good enough to get the point across but quiet small compare to modern models and applications. 1. Wang, Hongyi, et al. "Federated learning with matched averaging." ICLR 2020

Correctness: The methodogy followed on the paper seems to be adequate.

Clarity: The paper is very well written, cohesive, and easy to follow.

Relation to Prior Work: Paper mention most relevant related work, but I would like to see more details on how it is substancially different to FedMA

Reproducibility: Yes

Additional Feedback: This work have the potential of great impact among many applications. - I would like to get more details on the matching of layers of different size. Does the Wasserstein barycenter approah still applies in this setting? - I know you mention [1] on related work, but the work is so similar that it is still no obvious to the reader what's the difference beyond the approach taken to obtain the optimal transport. 1. Wang, Hongyi, et al. "Federated learning with matched averaging." ICLR 2020

[Author Response · NeurIPS 2020]

**R1**: **(a)** *"blur the distributions":* As Wasserstein barycenter *adjusts the support*, blurring is more likely for Euclidean
avg. **(b)** *continual learning:* Growing the barycentric network gradually & unbalanced OT is left for future work.

**R2**, **R4**: We present results on an additional (harder) dataset, CIFAR100, to illustrate that *our results indeed generalize!*
**1.** In Table 1, we adapt the VGG11 architecture (used for CIFAR10) and train multiple copies *with different*
*initializations*, in a similar manner for 300 epochs. Here, our focus was not to train individual models with best accuracy,
*rather to investigate the efficacy of fusion*. OT fusion results in a mean test accuracy gain $\sim \{1.4\%, 1.7\%, 2\%\}$ over
the best individual models, in case of $\{4, 6, 8\}-$ base models, and is *# model $\times$ more efficient* than ensembling them.
Vanilla averaging, in contrast, fails to fine-tune despite trying numerous settings of optimization hyperparameters. **2.**
Also, Fig 1, shows similar gains for *data-free post-processing* in case of structured pruning (as in Sec 5.2).

| CIFAR100 + VGG11 | INDIVIDUAL MODELS | PREDICTION AVG. | FINETUNING | |
| --- | --- | --- | --- | --- |
| | | | VANILLA | OT |
| Accuracy | [62.70, 62.57, 62.50, 62.92] | 66.32 | 4.02 | **64.29± 0.26** |
| Efficiency | | 1 × | 4 × | **4 ×** |
| Accuracy | [62.70, 62.57, 62.50, 62.92, 62.53, 62.70] | 66.99 | 0.85 | **64.55 ± 0.30** |
| Efficiency | | 1 × | 6 × | **6 ×** |
| Accuracy | [62.70, 62.57, 62.50, 62.92, 62.53, 62.70, 61.60, 63.20] | 67.28 | 1.00 | **65.05± 0.53** |
| Efficiency | | 1 × | 8 × | **8 ×** |

Table 1: Efficient alternative to ensembling via OT fusion on **CIFAR100** for VGG11. Vanilla average fails to retrain. Results shown are mean $\pm$ std. deviation over **5 seeds**.

Figure 1: Post-processing for structured pruning via OT-fusion on **CIFAR100**.

**R2**, **R3** *"there could possibly be more competent baselines":* **1.** We compare OT fusion in the context of: ensembling
(Sec 5.3), vanilla averaging (Sec 5.1, 5.3) widely used in federated learning, distillation (Sec 5.3, S12), & show a
favorable *accuracy-efficiency trade-off*. **2.** Averaging parameters of neural-networks with **different widths** (Sec 5.2) *is*
*being enabled for the first time*, to our knowledge. **3.** Greedily matching neurons performs worse than OT, as expected
theoretically.

**R2**: **(a)** *"forward for each of K individual models $\cdots$ compared to "prediction average"*: The activation-based alignment
(acts) does this **only once**, while prediction avg. will have to do this every time during inference. **(b)** *"published*
*structured pruning methods"*: Lines 295-297, our goal here is **not to propose a new method**, rather a post-processing
technique that is independent of the pruning algorithm. **(c)** We will surely organize the algorithm better.

**R3**: **(a)** *"special and general models $\cdots$ seems a bit artificial:* A similar setting was considered in the distillation paper
(Hinton et.al. 2015, Section 3), and likewise, in continual learning variants of this setup (Split-MNIST) are used for
benchmarking. The 'constraint' of performing this without sharing of sensitive training data arises in many applications,
such as healthcare, legal, etc. **(b)** *"improvement over vanilla averaging is very marginal"*: We respectfully disagree. **1.**
2-model case: Besides the results in Table 1 please refer to other fine-tuning settings in Table S7, S8 where OT fusion
also outperforms. Plus, we are fine-tuning for a significant duration ($\sim$ 100 epochs) to adequately illustrate that vanilla
avg. can't recover. **2.** $\geq 2$ models: Vanilla avg. fails to retrain despite trying a large set of hyperparameters (Appendix
S4.2), also check the results on CIFAR100 in Table 1, reported over 5 seeds. **(c)** *"people don't average the weights":*
As noted by **R2**, **R4**, and as discussed above, element-wise averaging of weights has a widespread adoption in federated
learning (FedAvg, McMahan et. al. 2016). **(d)** *Miscellaneous:* **1.** For structured pruning (Fig. 3), we use weight-based
variant to avoid the usage of data (Line 277). But, in general, activation-based alignment (acts) performs on par (and
often slightly better), so we use it for all other results (Line 193). **2.** Fig S9 caption: it should be "all".

**R3**, **R4**: *"model benefit from fusion with (almost) itself?"* Due to mass conservation when doing OT between dense and
pruned model layers, the (removed) filters of the dense model, which either detect similar features or whose features can
be composed, get fused into the remaining filters of the smaller model. We will add the activation maps in the paper.

**R4**: **(a)** *FedMA*. **1. Flexibility**: FedMA inherently solves a hard-assignment problem to obtain a permutation, while
our approach is based on the more general optimal transportation problem (OT). So, if the number of neurons being
matched are different, OT can transport a distribution [1/2, 1/2] to [1/4, 1/4, 1/4, 1/4 ] and vice versa. This fundamental
difference allows us to fuse into a smaller model (as illustrated by the two applications in Section 5.2), in a rather
effortless way using OT as compared to FedMA. **2. Practicality**: FedMA is restrictive from the practical viewpoint,
since it requires extensive coordination and communication. It assumes that same set of clients communicate repeatedly
**for # layer many rounds**, where each round involves freezing the previously matched layers across the devices, and
then matching the current layer. After which, the rest of the layers get retrained and the procedure is repeated until all
the layers get matched. But in practice (Kairouz et. al., 2019), the server samples a random subset of active devices in
each round. Also, straggler devices can **hinder a proper alignment** of models in FedMA, hence limiting its practical
applicability. **3. Stability:** Their intermittent "freezing and retraining" process is known to suffer from convergence
instabilities during retraining (see Appendix A of their paper). In contrast, our one-shot fusion of entire models via OT
does not suffer from these issues. **(b)** *matching of layers of different size:* The mass splitting example above should
better explain how the matching might behave (also see the shared point with **R3**).

[Meta-Review · NeurIPS 2020]

Although the reviewers had some concerns initially about baselines, small datasets, blurring the representations, etc., these were well addressed in the rebuttal and many of the reviewers have increased their scores. The consensus is that this approach will enable interesting future work on model fusion and potentially have a decent impact on areas like federated learning. Please include the requested clarifications and additional results in the final draft.